# ACCURATE BAYESIAN META-LEARNING BY ACCURATE TASK POSTERIOR INFERENCE

**Michael Volpp**[1,2][*]
**Philipp Dahlinger**[1]
**Philipp Becker**[1]
**Christian Daniel**[2]
**Gerhard Neumann**[1]

[1]Karlsruhe Institute of Technology, Karlsruhe, Germany
[2]Bosch Center for Artificial Intelligence, Renningen, Germany

## ABSTRACT

Bayesian meta-learning (BML) enables fitting expressive generative models to small datasets by incorporating inductive priors learned from a set of related tasks. The Neural Process (NP) is a prominent deep neural network-based BML architecture, which has shown remarkable results in recent years. In its standard formulation, the NP encodes epistemic uncertainty in an amortized, factorized, Gaussian variational (VI) approximation to the BML task posterior (TP), using reparametrized gradients. Prior work studies a range of architectural modifications to boost performance, such as attentive computation paths or improved context aggregation schemes, while the influence of the VI scheme remains under-explored. We aim to bridge this gap by introducing GMM-NP, a novel BML model, which builds on recent work that enables highly accurate, full-covariance Gaussian mixture (GMM) TP approximations by combining VI with natural gradients and trust regions. We show that GMM-NP yields tighter evidence lower bounds, which increases the efficiency of marginal likelihood optimization, leading to improved epistemic uncertainty estimation and accuracy. GMM-NP does not require complex architectural modifications, resulting in a powerful, yet conceptually simple BML model, which outperforms the state of the art on a range of challenging experiments, highlighting its applicability to settings where data is scarce.

## 1 INTRODUCTION

Driven by algorithmic advances in the field of deep learning (DL) and the availability of increasingly powerful GPU-assisted hardware, the field of machine learning achieved a plethora of impressive results in recent years (Parmar et al., 2018; Radford et al., 2019; Mnih et al., 2015). These were enabled to a large extent by the availability of huge datasets, which enables training expressive deep neural network (DNN) models. In practice, e.g., in industrial settings, such datasets are unfortunately rarely available, rendering standard DL approaches futile. Nevertheless, it is often the case that similar tasks arise repeatedly, such that the number of context examples on a novel target task is typically relatively small, but the joint *meta-dataset* of examples from all tasks accumulated over time can be massive, s.t. powerful inductive biases can be extracted using *meta-learning* (Hospedales et al., 2022). While these inductive biases allow restricting predictions to only those compatible with the meta-data, there typically remains *epistemic uncertainty* due to task ambiguity, as the context data is often not informative enough to identify the target task exactly. *Bayesian meta-learning* (BML) aims at an accurate quantification of this uncertainty, which is crucial for applications like active learning, Bayesian optimization (Shahriari et al., 2016), model-based reinforcement learning (Chua et al., 2018), robotics (Deisenroth et al., 2011), and in safety-critical scenarios.

---

[*]Correspondence to: `Michael.Volpp@de.bosch.com`

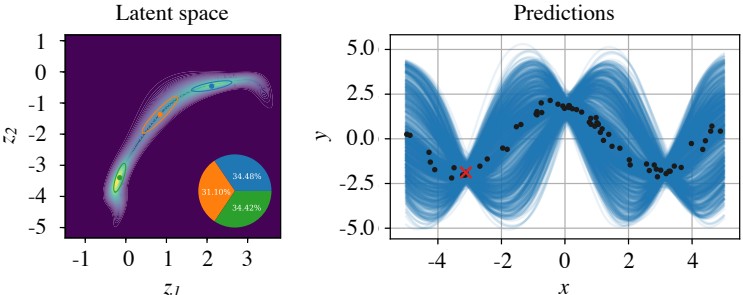

Figure 1: Visualization of our GMM-NP model for a $d_z = 2$ dimensional latent space, trained on a meta-dataset of sinusoidal functions with varying amplitudes and phases, after having observed a single context example (red cross, right panel) from an unseen task (black dots, right panel). Left panel: unnormalized task posterior (TP) distribution (contours) and GMM TP approximation with $K = 3$ components (ellipses, mixture weights in %). Right panel: corresponding function samples from our model (blue lines). A single context example leaves much task ambiguity, reflected in a highly correlated, multi-modal TP. Our GMM approximation correctly captures this: predictions are in accordance with (i) the observed data (all samples pass close to the red context example), and with (ii) the learned inductive biases (all samples are sinusoidal), cf. also Fig. 12 in App. A.5.5

A prominent BML approach is the Neural Process (NP) (Garnelo et al., 2018b) which employs a DNN-based conditional latent variable (CLV) model, in which the Bayesian belief about the target task is encoded in a factorized Gaussian task posterior (TP) approximation, and inference is amortized over tasks using set encoders (Zaheer et al., 2017). This architecture can be optimized efficiently using variational inference (VI) with standard, reparametrized gradients (Kingma & Welling, 2014). A range of modifications, such as adding deterministic, attentive, computation paths (Kim et al., 2019), or Bayesian set encoders (Volpp et al., 2021), have been proposed in recent years to improve predictive performance. Interestingly, the VI scheme with an amortized, factorized Gaussian TP, optimized using standard gradients, remains largely unaltered. Yet, it is well known that (i) the factorized Gaussian assumption rarely holds in Bayesian learning (MacKay, 2003; Wilson & Izmailov, 2020), (ii) amortized inference can yield suboptimal posterior approximations (Cremer et al., 2018), and (iii) natural gradients are superior to standard gradients for VI in terms of optimization efficiency and robustness (Khan & Nielsen, 2018).

Building on these insights and on recent advances in VI (Lin et al., 2020; Arenz et al., 2022), we propose GMM-NP, a novel NP-based BML algorithm that employs (i) a full-covariance Gaussian mixture (GMM) TP approximation, optimized in a (ii) non-amortized fashion, using (iii) robust and efficient trust region natural gradient (TRNG)-VI. We demonstrate through extensive empirical evaluations and ablations that our approach yields tighter evidence lower bounds, more efficient model optimization, and, thus, markedly improved predictive performance, outperforming the state-of-the-art both in terms of epistemic uncertainty quantification and accuracy. Notably, GMM-NP does not require complex architectural modifications, which shows that accurate TP inference is crucial for accurate BML, an insight we believe will be valuable for future research.

## 2 RELATED WORK

Multi-task learning aims to leverage inductive biases learned on a meta-dataset of similar tasks for improved data efficiency on unseen target tasks of similar structure. Notable variants include transfer-learning (Zhuang et al., 2020), that refines and combines pre-trained models (Golovin et al., 2017; Krizhevsky et al., 2012), and meta-learning (Schmidhuber, 1987; Thrun & Pratt, 1998; Vilalta & Drissi, 2005; Hospedales et al., 2022), which makes the multi-task setting explicit in the model design by formulating fast adapation mechanisms in order to learn *how* to solve tasks with little context data ("few-shot learning"). A plethora of architectures were studied in the literature, including learner networks that adapt model parameters (Bengio et al., 1991; Schmidhuber, 1992; Ravi & Larochelle, 2017), memory-augmented DNNs (Santoro et al., 2016), early instances of Bayesian meta-models (Edwards & Storkey, 2017; Hewitt et al., 2018), and algorithms that that make use of learned measures of task similarity (Koch et al., 2015; Vinyals et al., 2016; Snell et al., 2017).

Arguably the most prominent meta-learning approaches are the Model-agnostic Meta-learning (MAML) and the Neural Process (NP) model families, due to their generality and flexibility. While the original MAML (Finn et al., 2017) and Conditional NP (Garnelo et al., 2018a) formulations do

not explicitly model the epistemic uncertainty arising naturally in few-shot settings due to task ambiguity, both model families were extended to fully Bayesian meta-learning (BML) algorithms that explicitly infer the TP based on a CLV formulation (Heskes, 2000; Bakker & Heskes, 2003). Important representatives are Probabilistic MAML (Grant et al., 2018; Finn et al., 2018) and Bayesian MAML (Kim et al., 2018), as well as several NP-based BML approaches that inspire our work. These include the Standard NP (Garnelo et al., 2018b), which was extended by attentive computation paths to avoid underfitting (Kim et al., 2019), or by Bayesian set encoders (Zaheer et al., 2017; Wagstaff et al., 2019; Volpp et al., 2020) for improved handling of task ambiguity, as well as by hierarchical (Wang & Van Hoof, 2020), bootstrapped (Lee et al., 2020), or graph-based (Louizos et al., 2019) latent distributions. While the original NP formulation employs an amortized, reparametrized, stochastic gradient VI objective (Kingma & Welling, 2014; Rezende et al., 2014), Monte-Carlo (MC)-based objective functions were also studied (Gordon et al., 2019; Volpp et al., 2021).

From a more general perspective, VI emerged as a central tool in many areas of probabilistic machine learning, which require tractable approximations of intractable probability distributions, typically arising as the posterior in Bayesian models (Gelman et al., 2004; Koller & Friedman, 2009; Neal, 1996; Wilson & Izmailov, 2020). While early approaches (Attias, 2000) allow analytic updates, more complex algorithms employ stochastic gradients w.r.t. the variational parameters (Ranganath et al., 2014; Kingma & Welling, 2014; Blundell et al., 2015). Such approaches are straightforward to implement and computationally efficient for factorized Gaussian variational distributions, but ignore the information geometry of the loss landscape, leading to suboptimal convergence rates (Khan & Nielsen, 2018). Natural gradient (NG)-VI (Amari, 1998) alleviates this problem and recent work (Hoffman et al., 2013; Winn & Bishop, 2005; Khan & Nielsen, 2018; Khan et al., 2018) successfully applies this idea at scale to complex models, requiring only first-order gradient information (Lin et al., 2019). Further extensions enable NG-VI for structured variational distributions such as mixture models by decomposing the NG update into individual updates per mixture component (Arenz et al., 2018; Lin et al., 2020) which, in combination with trust region (TR) step size control (Abdolmaleki et al., 2015; Arenz et al., 2022), yields robust and efficient VI algorithms for versatile and highly expressive variational distributions such as Gaussian mixture models (GMMs).

## 3 PRELIMINARIES

We now briefly recap the TRNG-VI algorithm (Lin et al., 2020; Arenz et al., 2022) as well as the NP model (Garnelo et al., 2018b), which form the central building blocks of our GMM-NP model.

### 3.1 TRUST REGION NATURAL GRADIENT VI WITH GAUSSIAN MIXTURE MODELS

**Variational Inference.** We consider a probability distribution $p(z)$ over a random variable $z \in \mathbb{R}^{d_z}$, which is intractable in the sense that we know it only up to some normalization constant $Z$, i.e., $p(z) = \tilde{p}(z) / Z$ with $Z = \int p(z)\, \mathrm{d}z$ and tractable $\tilde{p}(z)$. We seek to approximate $p(z)$ by a tractable distribution $q_\phi(z)$, parametrized by $\phi$. Variational inference (VI) frames this task as the minimization w.r.t. $\phi$ of the reverse Kullback-Leiber (KL) divergence (Kullback & Leibler, 1951)

$$\mathrm{KL}\left[q_\phi \| p\right] \equiv -\mathbb{E}_{q_\phi(z)}\left[\log \frac{\tilde{p}(z)}{q_\phi(z)}\right] + \log Z \equiv -\mathcal{L}(\phi) + \log Z, \tag{1}$$

where we introduced evidence lower bound (ELBO) $\mathcal{L}(\phi)$. As $Z$ is independent of $\phi$, minimizing the KL divergence is equivalent to maximizing the ELBO.

**Natural Gradients.** A standard approach employs stochastic, reparametrized gradients w.r.t. $\phi$ (Kingma & Welling, 2014) for optimization. While this is computationally efficient, it ignores the geometry of the statistical manifold defined by the set of probability distributions $q_\phi$, which can lead to suboptimal convergence rates (Khan & Nielsen, 2018). A more efficient solution is to perform updates in the *natural gradient* (NG) direction, i.e., the direction of steepest ascent w.r.t. the Fisher information metric (Amari, 1998). State-of-the-art approaches estimate the NG from first-order gradients of $p(z)$ by virtue of Stein's lemma (Lin et al., 2019), yielding efficient NG-VI algorithms that scale to complex problems (Khan et al., 2018; Lin et al., 2020; Arenz et al., 2022).

**Trust Regions.** Selecting appropriate step sizes for updates in $\phi$ can be intricate, which is why Abdolmaleki et al. (2015) propose a (zero-order) algorithm that incorporates a *trust region* constraint

of the form $\mathrm{KL}\left[q_{\boldsymbol{\phi}} \| q_{\boldsymbol{\phi}_{\mathrm{old}}}\right] \leq \varepsilon$, which restricts the updates in distribution space and can be enforced with manageable computational overhead (a scalar, convex optimization problem in the Lagrangean parameter for the constraint). As shown by Arenz et al. (2022), such trust regions can easily be combined with gradient information, and allow more aggressive updates in comparison to setting the step size directly, while still ensuring robust convergence.

**VI with Gaussian Mixture Models.** The quality of the approximation depends on the expressiveness of the distribution family $q_{\boldsymbol{\phi}}$. In settings where $p$ corresponds to the Bayesian posterior of complex latent variable models (MacKay, 2003; Wilson & Izmailov, 2020), simple Gaussian approximations do not yield satisfactory results, as $p$ typically is multimodal. In such cases, Gaussian mixture models (GMMs) are an appealing choice, as they provide cheap sampling, evaluation, and marginalization while allowing expressive approximations (Arenz et al., 2018). However, a naive application of VI is futile because gradients are coupled between GMM components, leading to computationally intractable updates. Fortunately, Arenz et al. (2018) and Lin et al. (2020) show that updating the components and weights individually is possible, while preventing a collapse of the approximation onto a single posterior mode. This leads to two state-of-the-art algorithms for NG-VI with variational GMMs, that differ most notably in the way the step sizes for the updates are controlled: iBayes-GMM (Lin et al., 2020), which directly sets step sizes for the updates, and TRNG-VI (Arenz et al., 2022), which employs trust regions for more efficient and robust convergence.

## 3.2 BAYESIAN META-LEARNING WITH NEURAL PROCESSES

**The Multi-Task Latent Variable Model.** We aim to fit a generative model to a meta-dataset $\mathcal{D} = \mathcal{D}_{1:L}$, consisting of regression *tasks* $\mathcal{D}_{\ell} = \{\boldsymbol{x}_{\ell,1:N}, \boldsymbol{y}_{\ell,1:N}\}$ with inputs $\boldsymbol{x}_{\ell,n} \in \mathbb{R}^{d_x}$ and corresponding evaluations $\boldsymbol{y}_{\ell,n} \in \mathbb{R}^{d_y}$ of unknown functions $f_{\ell}$, i.e., $\boldsymbol{y}_{\ell,n} = f_{\ell}(\boldsymbol{x}_{\ell,n}) + \boldsymbol{\varepsilon}_n$, where $\boldsymbol{\varepsilon}_n$ denotes (possibly heteroskedastic) noise. Tasks are assumed to share statistical structure as formalized in the multi-task CLV model shown to the right, defining the joint probability distribution

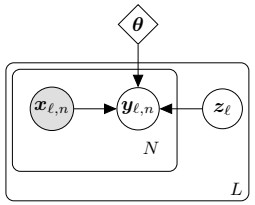

$$p_{\boldsymbol{\theta}}\left(\boldsymbol{y}_{1:L,1:N}, \boldsymbol{z}_{1:L} | \boldsymbol{x}_{1:L,1:N}\right) = \prod_{\ell,n} p_{\boldsymbol{\theta}}\left(\boldsymbol{y}_{\ell,n} | \boldsymbol{x}_{\ell,n}, \boldsymbol{z}_{\ell}\right) p\left(\boldsymbol{z}_{\ell}\right), \tag{2}$$

where $\boldsymbol{z}_{\ell} \in \mathbb{R}^{d_z}$ denote latent task descriptors and $\boldsymbol{\theta}$ denotes task-global parameters that capture shared statistical structure. Having observed context data $\mathcal{D}_*^c = \{\boldsymbol{x}_{*,1:M_*}^c, \boldsymbol{y}_{*,1:M_*}^c\} \subset \mathcal{D}_*$ from a target task $\mathcal{D}_*$, predictions are provided in terms of the marginal predictive distribution

$$p_{\boldsymbol{\theta}}\left(\boldsymbol{y}_{*,1:N_*} | \boldsymbol{x}_{*,1:N_*}, \mathcal{D}_*^c\right) = \int \prod_n p_{\boldsymbol{\theta}}\left(\boldsymbol{y}_{*,n} | \boldsymbol{x}_{*,n}, \boldsymbol{z}_*\right) p_{\boldsymbol{\theta}}\left(\boldsymbol{z}_* | \mathcal{D}_*^c\right) \mathrm{d}\boldsymbol{z}_*, \tag{3}$$

with the *task posterior* (TP) distribution $p_{\boldsymbol{\theta}}\left(\boldsymbol{z}_* | \mathcal{D}_*^c\right) = \prod_m p_{\boldsymbol{\theta}}(\boldsymbol{y}_{*,m}^c | \boldsymbol{x}_{*,m}^c, \boldsymbol{z}_*) p\left(\boldsymbol{z}_*\right) / p_{\boldsymbol{\theta}}\left(\mathcal{D}_*^c\right)$.

**The Neural Process.** In its standard formulation, the Neural Process (NP) (Garnelo et al., 2018b) defines a factorized Gaussian likelihood $p_{\boldsymbol{\theta}}\left(\boldsymbol{y} | \boldsymbol{x}, \boldsymbol{z}\right) \equiv \mathcal{N}\left(\boldsymbol{y} | \mathrm{dec}_{\boldsymbol{\theta}}^{\boldsymbol{\mu}}\left(\boldsymbol{x}, \boldsymbol{z}\right), \mathrm{diag}\left(\sigma_{\mathrm{n}}^2\right)\right)$, where the *decoder* $\mathrm{dec}_{\boldsymbol{\theta}}^{\boldsymbol{\mu}}$ is a DNN with weights $\boldsymbol{\theta}$, and observation noise variance $\sigma_{\mathrm{n}}^2$. As the TP is intractable for this likelihood choice, NP computes a factorized Gaussian approximation $q_{\boldsymbol{\phi}}\left(\boldsymbol{z}_* | \mathcal{D}_*^c\right) \equiv \mathcal{N}\left(\boldsymbol{z}_* | \mathrm{enc}_{\boldsymbol{\phi}}^{\boldsymbol{\mu}}\left(\mathcal{D}_*^c\right), \mathrm{diag}(\mathrm{enc}_{\boldsymbol{\phi}}^{\boldsymbol{\sigma}}\left(\mathcal{D}_*^c\right))\right)$ with *deep set encoders* (Zaheer et al., 2017; Wagstaff et al., 2019) $\mathrm{enc}_{\boldsymbol{\phi}}^{\boldsymbol{\mu}}$, $\mathrm{enc}_{\boldsymbol{\phi}}^{\boldsymbol{\sigma}}$, parametrized by $\boldsymbol{\phi}$. The parameters $\boldsymbol{\Phi} \equiv (\boldsymbol{\theta}, \boldsymbol{\phi})$ are optimized jointly on the meta-data by stochastic gradient ascent on the ELBO $\sum_{\ell=1}^L \mathcal{L}_{\ell}(\boldsymbol{\Phi})$ w.r.t. the approximate log marginal predictive likelihood defined by

$$\log q_{\boldsymbol{\Phi}}\left(\boldsymbol{y}_{\ell,1:N} | \boldsymbol{x}_{\ell,1:N}, \mathcal{D}_{\ell}^c\right) \equiv \log \int \prod_n p_{\boldsymbol{\theta}}\left(\boldsymbol{y}_{\ell,n} | \boldsymbol{x}_{\ell,n}, \boldsymbol{z}_{\ell}\right) q_{\boldsymbol{\phi}}\left(\boldsymbol{z}_{\ell} | \mathcal{D}_{\ell}^c\right) \mathrm{d}\boldsymbol{z}_{\ell} \tag{4}$$

$$\geq \mathbb{E}_{q_{\boldsymbol{\phi}}(\boldsymbol{z}_{\ell} | \mathcal{D}_{\ell})}\left[\sum_n \log p_{\boldsymbol{\theta}}\left(\boldsymbol{y}_{\ell,n} | \boldsymbol{x}_{\ell,n}, \boldsymbol{z}_{\ell}\right) + \log \frac{q_{\boldsymbol{\phi}}\left(\boldsymbol{z}_{\ell} | \mathcal{D}_{\ell}^c\right)}{q_{\boldsymbol{\phi}}\left(\boldsymbol{z}_{\ell} | \mathcal{D}_{\ell}\right)}\right] \equiv \mathcal{L}_{\ell}(\boldsymbol{\Phi}), \tag{5}$$

where $\mathcal{D}_{\ell}^c \subset \mathcal{D}_{\ell}$, and stochastic gradients w.r.t. $\boldsymbol{\phi}$ are estimated using the reparametrization trick (Kingma & Welling, 2014). Note that NP *amortizes* inference (the variational parameters $\boldsymbol{\phi}$ are shared across tasks) and that it re-uses $q_{\boldsymbol{\phi}}\left(\boldsymbol{z}_{\ell} | \cdot\right)$ to compute the variational distribution $q_{\boldsymbol{\phi}}\left(\boldsymbol{z}_{\ell} | \mathcal{D}_{\ell}\right)$, taking advantage of its deep set encoder, which allows to condition it on datasets of arbitrary size.

## 4    BAYESIAN META-LEARNING WITH GMM TASK POSTERIORS

**Motivation.** Our work is motivated by the observation that the current state-of-the-art approach for training NP-based BML models is suboptimal. Concretely, we identify three interrelated issues with the optimization objective Eq. (5):

(I1) **Expressivity of the Variational Distribution.** $q_\phi$ is a (i) factorized, (ii) unimodal Gaussian distribution, (iii) amortized over tasks. In effect, this parametrization only allows crude approximations of the TP distribution (MacKay, 2003; Cremer et al., 2018).

(I2) **Optimization of the Variational Parameters.** (i) Naive gradients of Eq. (5), ignoring the information geometry of $q_\phi$, with (ii) direct step size control are employed for optimization, yielding brittle convergence at suboptimal rates (Khan & Nielsen, 2018; Arenz et al., 2022).

(I3) **Optimization of the Model Parameters.** Due to the suboptimal VI scheme (I1,I2), the TP approximation is poor, resulting in a loose ELBO Eq. (5). In effect, optimization w.r.t. the model parameters $\boldsymbol{\theta}$ is inefficient, cf. App. A.1.4 for a detailed discussion.

Armed with these insights, we develop a novel BML model algorithm that is close in spirit to the NP but solves (I1-I3) through TRNG-VI with GMM TP approximations.

**Model.** Our algorithm builds on the standard multi-task CLV architecture Eq. (2) and retains the likelihood parametrization using a decoder DNN, $\mathrm{dec}_{\boldsymbol{\theta}}^{\boldsymbol{\mu}}(\boldsymbol{x}, \boldsymbol{z})$, as this allows for expressive BML models. Under this parametrization, the log marginal likelihood for a single task reads

$$\log p_{\boldsymbol{\theta}}\left(\boldsymbol{y}_{\ell,1:N}|\boldsymbol{x}_{\ell,1:N}\right) = \log \int \prod_n p_{\boldsymbol{\theta}}\left(\boldsymbol{y}_{\ell,n}|\boldsymbol{x}_{\ell,n}, \boldsymbol{z}_\ell\right) p\left(\boldsymbol{z}_\ell\right) \mathrm{d}\boldsymbol{z}_\ell \equiv \log Z_\ell\left(\boldsymbol{\theta}\right), \qquad (6)$$

where $Z_\ell\left(\boldsymbol{\theta}\right)$ is the normalization constant of the TP $p_{\boldsymbol{\theta}}\left(\boldsymbol{z}_\ell|\mathcal{D}_\ell\right) = \tilde{p}_\ell\left(\boldsymbol{z}_\ell\right)/Z_\ell\left(\boldsymbol{\theta}\right)$ with $\tilde{p}(\boldsymbol{z}_\ell) \equiv \prod_n p_{\boldsymbol{\theta}}\left(\boldsymbol{y}_{\ell,n}|\boldsymbol{x}_{\ell,n}, \boldsymbol{z}_\ell\right) p\left(\boldsymbol{z}_\ell\right)$. In contrast to Eq. (4), we do not condition the left hand side on a context set $\mathcal{D}_\ell^c$, which yields a tractable integrand $\tilde{p}(\boldsymbol{z}_\ell)$ that does not require further approximation. To tackle (I1), we approximate $p_{\boldsymbol{\theta}}\left(\boldsymbol{z}_\ell|\mathcal{D}_\ell\right)$ by an expressive variational GMM of the form

$$q_{\boldsymbol{\phi}_\ell}\left(\boldsymbol{z}_\ell\right) \equiv \sum_k w_{\ell,k} q_{\boldsymbol{\phi}_\ell}\left(\boldsymbol{z}_\ell|k\right) \equiv \sum_k w_{\ell,k} \mathcal{N}\left(\boldsymbol{z}_\ell|\boldsymbol{\mu}_{\ell,k}, \boldsymbol{\Sigma}_{\ell,k}\right), \quad \sum_k w_{\ell,k} = 1, \qquad (7)$$

where we train individual GMMs with parameters $\boldsymbol{\phi}_\ell \equiv \{w_{\ell,k}, \boldsymbol{\mu}_{\ell,k}, \boldsymbol{\Sigma}_{\ell,k}\}$, $k \in \{1, \ldots, K\}$ for each task $\ell$, to not impair approximation quality by introducing inaccuracies through amortization.

**Update Equations for the Variational Parameters.** To ensure efficient and robust optimization of $\boldsymbol{\phi}_\ell$ (I2), we employ TRNG-VI as proposed by Arenz et al. (2022), with the update equations

$$\boldsymbol{\Sigma}_{\ell,k,\mathrm{new}} = \left[\frac{\eta}{\eta+1}\boldsymbol{\Sigma}_{\ell,k,\mathrm{old}}^{-1} - \frac{1}{\eta+1}\boldsymbol{R}_{\ell,k}\right]^{-1}, \qquad (8\mathrm{a})$$

$$\boldsymbol{\mu}_{\ell,k,\mathrm{new}} = \boldsymbol{\Sigma}_{\ell,k,\mathrm{new}}\left[\frac{\eta}{\eta+1}\boldsymbol{\Sigma}_{\ell,k,\mathrm{old}}^{-1}\boldsymbol{\mu}_{\ell,k,\mathrm{old}} + \frac{1}{\eta+1}\left(\boldsymbol{r}_{\ell,k} - \boldsymbol{R}_{\ell,k}\boldsymbol{\mu}_{\ell,k,\mathrm{old}}\right)\right], \qquad (8\mathrm{b})$$

$$w_{\ell,k,\mathrm{new}} \propto \exp \rho_{\ell,k}, \qquad (8\mathrm{c})$$

where $\boldsymbol{R}_{\ell,k}$, $\boldsymbol{r}_{\ell,k}$, and $\rho_{\ell,k}$ are defined as expectations that can be approximated from per-component samples using MC and require at most first-order gradients of $\tilde{p}(\boldsymbol{z}_\ell)$, which are readily available using standard automatic differentiation software (Abadi et al., 2015; Paszke et al., 2019). Due to space constraints, we move details to App. A.1.1. The optimal value for the trust region parameter $\eta \geq 0$ is defined by a scalar convex optimization problem that can be solved efficiently by a bracketing search, which also ensures positive definiteness of the new covariance matrix $\boldsymbol{\Sigma}_{\ell,k,\mathrm{new}}$.

**Updates for the Model Parameters.** To optimize the model parameters $\boldsymbol{\theta}$, we decompose the log marginal likelihood $\log Z_\ell\left(\boldsymbol{\theta}\right)$ according to Eq. (1) as

$$\log Z_\ell\left(\boldsymbol{\theta}\right) = \mathbb{E}_{q_{\boldsymbol{\phi}_\ell}(\boldsymbol{z}_\ell)}\left[\sum_n \log p_{\boldsymbol{\theta}}\left(\boldsymbol{y}_{\ell,n}|\boldsymbol{x}_{\ell,n}, \boldsymbol{z}_\ell\right) + \log \frac{p\left(\boldsymbol{z}_\ell\right)}{q_{\boldsymbol{\phi}_\ell}\left(\boldsymbol{z}_\ell|\mathcal{D}_\ell\right)}\right] + \mathrm{KL}\left[q_{\boldsymbol{\phi}_\ell}\left(\cdot\right) \| p_{\boldsymbol{\theta}}\left(\cdot|\mathcal{D}_\ell\right)\right],$$
$$(9)$$

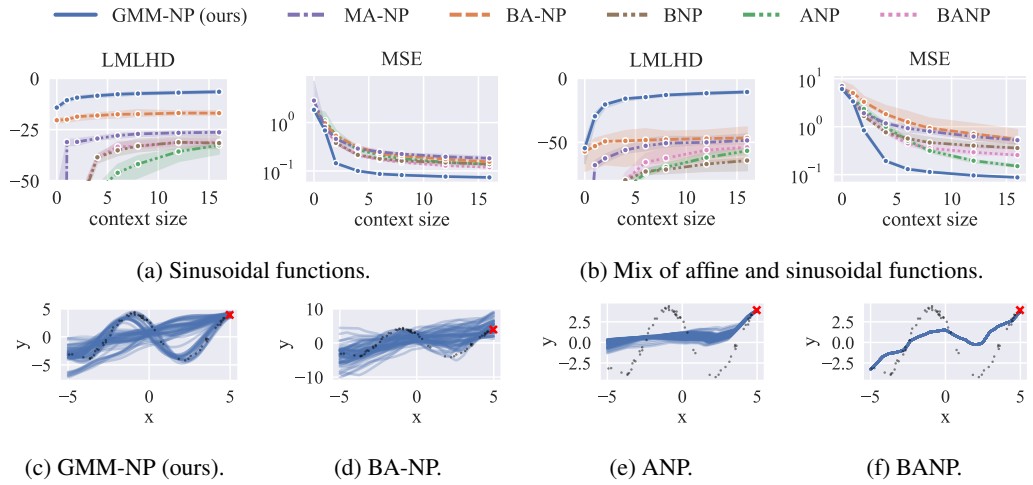

Figure 2: Panels (a), (b): LMLHD and MSE on two synthetic function classes. Panels (c) – (f): function samples of models trained on the affine-sinusoidal class (b), given one context example (red) from a sinusoidal instance (black). GMM-NP outperforms the baselines, as it accurately quantifies epistemic uncertainty through diverse samples. BA-NP also shows variability in its samples, but does not achieve competitive performance due to its inaccurate TP approximation. ANP and BANP produce essentially deterministic predictions that fail to give reasonable estimates of the predictive distribution. Cf. also Figs. 10, 11 in App. A.5.4.

where the first term on the right hand side is the ELBO w.r.t. $\log Z_\ell(\boldsymbol{\theta})$, which we denote by $\mathcal{L}(\boldsymbol{\theta})$. We expect $\mathcal{L}(\boldsymbol{\theta})$ to be comparably tight, as our inference scheme allows accurate GMM TP approximations $q_{\boldsymbol{\phi}_\ell}$, s.t., the KL term will be small. Consequently, maximization of $Z_\ell(\boldsymbol{\theta})$ w.r.t. $\boldsymbol{\theta}$ can be performed efficiently by maximization of $\mathcal{L}(\boldsymbol{\theta})$ (I3), cf. also App. A.1.4. As is standard, we use the Adam optimizer (Kingma & Ba, 2015) to perform updates in $\boldsymbol{\theta}$, with MC gradient estimates from samples $\boldsymbol{z}_{\ell,s} \sim q_{\boldsymbol{\phi}_\ell}(\boldsymbol{z}_\ell)$: $\nabla_{\boldsymbol{\theta}} \mathcal{L}(\boldsymbol{\theta}) \propto \sum_{s,n} \nabla_{\boldsymbol{\theta}} \log p_{\boldsymbol{\theta}}(\boldsymbol{y}_{\ell,n}|\boldsymbol{x}_{\ell,n}, \boldsymbol{z}_{\ell,s})$.

**Meta-Training.** The goal of any BML algorithm is to compute accurate predictions with well-calibrated uncertainty estimates according to Eq. (4), based on samples from the approximate TP $q_{\boldsymbol{\phi}_*}(\boldsymbol{z}_*|\mathcal{D}_*^c) \approx p_{\boldsymbol{\theta}}(\boldsymbol{z}|\mathcal{D}_*^c)$, conditioned on a context set $\mathcal{D}_*^c$ from a target task. During a meta-training stage on meta-data $\mathcal{D}_{1:L}$, we aim to encode inductive biases in the model parameters $\boldsymbol{\theta}$, s.t. small (few-shot) context sets $\mathcal{D}_*^c$ suffice for accurate predictions. To find versatile solutions that work for variable context set sizes, it is necessary to emulate this during meta-training by evaluating gradients for $\boldsymbol{\theta}$ on samples $\boldsymbol{z}_{\ell,s}$ from approximate TPs informed by a range of context set sizes. Standard NPs achieve this by sampling a minibatch of auxiliary subtasks, with a random number of datapoints, from $\mathcal{D}_{1:L}$ for each step in the parameters $\boldsymbol{\Phi}$ (cf. Sec. A.3.2). Our algorithm uses a similar approach: starting from a fixed set of randomly initialized variational GMMs $\boldsymbol{\phi}_\ell$, and a randomly initialized model $\boldsymbol{\theta}$, we iterate through the meta-data in minibatches of auxiliary subtasks, and perform one update step in $\boldsymbol{\phi}_\ell$ for all subtasks in the minibatch, according to Eqs. (8), followed by one gradient step in $\boldsymbol{\theta}$. Thus, variational and model parameters evolve jointly in a similar fashion as for standard NP, resulting in a meta-training stage with comparable computational complexity, cf. App. A.5.6. As this approach retains a fixed set of variational GMMs over the whole course of meta-training (one for each auxiliary subtask), we accordingly sample a fixed set of auxiliary subtasks at the beginning of meta-training. We summarize our algorithm in App. A.1, Alg. 1.

**Predictions.** As our architecture does not amortize inference over tasks and, thus, does not learn a set encoder architecture, the variational GMMs learned during meta-training are not required for predictions on test tasks and can be discarded. To make predictions, we fix the model parameters $\boldsymbol{\theta}$ and fit a new variational GMM $q_{\boldsymbol{\phi}_*}$ to $\mathcal{D}_*^c$ by iterating Eqs. (8) until convergence. Afterwards, we can cheaply generate arbitrarily many samples $\boldsymbol{z}_{*,s} \sim q_{\boldsymbol{\phi}_*}(\boldsymbol{z}_*)$, and generate corresponding function samples, evaluated at arbitrary input locations $\boldsymbol{x}_*$, by a single forward pass through the decoder DNN to approximate the predictive distribution according to Eq. (4), cf. App. A.1, Alg. 2.

## 5 EMPIRICAL EVALUATION

Our empirical evaluation aims to study the effect on the predictive performance of (i) our improved TRNG-VI approach as well as of (ii) expressive variational GMM TP approximations in NP-based

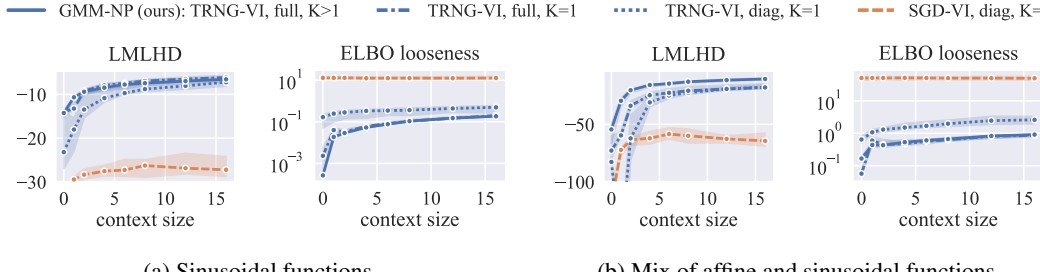

(a) Sinusoidal functions.

(b) Mix of affine and sinusoidal functions.

Figure 3: LMLHD and ELBO looseness over context set size for different versions of our algorithm (blue). Our improved TRNG-VI inference scheme yields tighter ELBOs than standard SGD-VI (orange) and, thus, improved performance (cf. text and App. A.1.4 for details).

BML, in (iii) comparison to the state-of-the-art on (iv) a range of practically relevant meta-learning tasks. To this end, we evaluate our GMM-NP architecture on a diverse set of BML experiments, and present comparisons to state-of-the-art BML algorithms, namely the original NP with mean context aggregation (MA-NP) (Garnelo et al., 2018b), the NP with Bayesian context aggregation (BA-NP) (Volpp et al., 2021), the Attentive NP (ANP) (Kim et al., 2019), as well as the Bootstrapping (Attentive) NP (B(A)NP) (Lee et al., 2020). Tab. 1 in App. A.2 gives an overview of the architectural differences of these algorithms. We move details on data generation to App. A.4, and on the baseline implementations to App. A.2. For a fair comparison, we employ a fixed experimental protocol for all datasets and models: we first perform a Bayesian hyperparameter search (HPO) to determine optimal algorithm settings, individually for each model-dataset combination. We then retrain the best model with 8 different random seeds and report the median log marginal predictive likelihood (LMLHD) as well as the median mean squared error (MSE), both in dependence of the context set size. To foster reproducibility, we provide further details on our experimental protocol in App. A.3, the resulting hyperparameters and architecture sizes in App. A.5.7, and publish our source code.[1] Lastly, we include a detailed discussion of limitations and computational resources in App. A.5.6.

## 5.1 SYNTHETIC DATASETS

We first study two synthetic function classes (Finn et al., 2017; 2018) on which predictions can be easily visualized: (i) sinusoidal functions with varying amplitudes and phases, as well as (ii) a mix of these sinusoidal functions with affine functions with varying slopes and intercepts. Fig. 2 shows that our GMM-NP outperforms all baselines by a large margin over the whole range of context sizes, both in terms of LMLHD and MSE. This indicates that GMM-NP's improved TP approximation indeed yields improved epistemic uncertainty estimation (higher LMLHD). Interestingly, GMM-NP also shows improved accuracy (lower MSE) and, notably, achieves this without any additional architectural modifications like parallel deterministic paths with attention modules. This is particularly pleasing, as the results show that such deterministic paths indeed improve accuracy, but degrade epistemic uncertainty estimation massively: (B)ANP performs worst in terms of LMLHD. This is further substantiated by (i) observing that MA-NP and BA-NP, both of which don't employ deterministic paths, are among the best baselines w.r.t. LMLHD, and (ii) by visualizing model predictions (Figs. 2, 10, 11), demonstrating that (B)ANP compute essentially deterministic function samples that fail to correctly estimate the predictive distribution, while our GMM-NP yields estimates uncertainty well through variable samples. BNP does not achieve competitive performance, presumably because the bootstrapping approach does not work well for small context sets.

## 5.2 ABLATION: TASK POSTERIOR INFERENCE

We now demonstrate that GMM-NP's improved performance can indeed be explained by the improved TRNG-VI algorithm with accurate GMM TP approximation. To this end, we compare: (i) BA-NP, i.e., amortized VI with reparameterized gradients and unimodal, factorized Gaussian TP (SGD-VI, diag, $K = 1$), (ii) our GMM-NP, i.e., non-amortized TRNG-VI and full-covariance GMM TP (TRNG-VI, full, $K > 1$), as well as two models employing TRNG-VI, but a unimodal Gaussian TP with (iii) full, and (iv) diagonal covariance. The results are shown in Fig. 3. In addition, we compare (v) an architecture with full-covariance GMM TP, but trained with iBayes-GMM (Lin et al., 2020), i.e., with direct step size control instead of trust regions (Fig. 6, App. A.5.1).

---

[1] https://github.com/ALRhub/gmm_np

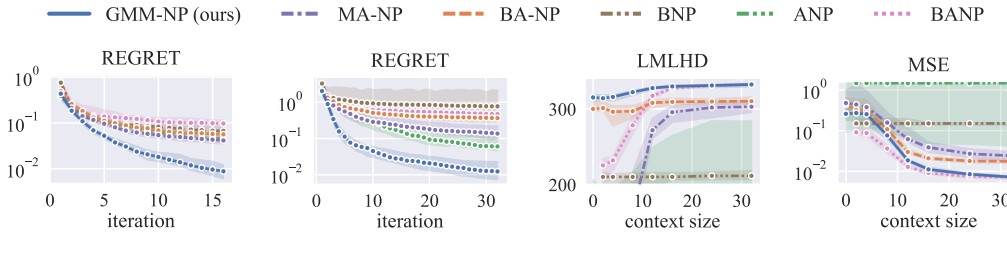

(a) BO on Branin (2D).  (b) BO on Hartmann (3D).  (c) 4D Furuta dynamics prediction.

Figure 4: Panels (a), (b): simple regret over iteration, when using BML models as Bayesian optimization (BO) surrogates (further results in App. A.5.2). As BO relies on well-calibrated uncertainty predictions, the results demonstrate that GMM-NP provides superior uncertainty estimates. Panel (c): log marginal likelihood (LMLHD) and MSE on one-step ahead predictions of 4D Furuta pendulum dynamics. While GMM-NP generally performs best, BANP also shows strong results.

**VI Algorithm.** Considering the LMLHD metric, we observe a significant performance boost when keeping the traditional factorized Gaussian approximation, but switching from SGD-VI to TRNG-VI, indicating that the standard SGD-VI approach is indeed suboptimal for BML. To study this further, we estimate the looseness of the ELBO (cf. App. A.3.3), i.e., the median (over tasks) value of the KL divergence $\mathrm{KL}\left[q_{\phi_\ell}\left(\cdot|\mathcal{D}_\ell\right) \| p_\theta\left(\cdot|\mathcal{D}_\ell\right)\right]$ between the true and approximate TPs. We observe that TRNG-VI provides ELBOs that are tighter by at least one order of magnitude in comparison to SGD-VI. As discussed above, this allows for more efficient optimization of the model parameters $\theta$, explaining the performance gain. Lastly, we find that trust regions yield tighter ELBOs than direct step size control and, consequently, improve predictive performance, cf. Fig. 6, App. A.5.1

**Posterior Expressivity.** We now study the effect of increasing the expressiveness of the TP approximation. This discussion is supplemented by Fig. 1, where we visualize the TP and its approximation for a $d_z = 2$ dimensional latent space. First, we observe tighter ELBOs and improved performance when considering full-covariance (but still unimodal) Gaussian TP approximations, and this effect is particularly pronounced for small context sets. This is intuitive, as small context sizes leave a lot of task ambiguity, leading to highly correlated latent dimensions (Fig. 1). If we now switch to multimodal TP approximations, i.e., our full GMM-NP architecture with $K > 1$ components ($K$ optimized by HPO), we observe a further increase in performance, as the multimodality of the true TP can be captured more accurately (Figs. 1,12). This effect is especially pronounced for the affine-sinusoidal mix, but also present for the purely sinusoidal function class. As more complex function classes exhibit stronger task ambiguity, the TP will likely exhibit multimodal, correlated structure over wider ranges of context sizes, s.t. an accurate TP approximation will be even more important.

## 5.3 BAYESIAN OPTIMIZATION

One important application area for probabilistic regression models is as the surrogate model of Bayesian optimization (BO), a global black-box optimization algorithm well-known for its sampling efficiency (Shahriari et al., 2016). BO serves as an interesting experiment to benchmark Bayesian models, as it relies on well-calibrated uncertainty estimates in order to trade-off exploration against exploitation, which is crucial for efficient optimization. As proposed by Garnelo et al. (2018b), we use Thompson sampling (Russo et al., 2018) as the BO acquisition function and present results on four function classes: (i) 1D functions sampled from Gaussian process (GP) priors with RBF kernels with varying lengthscales and signal variances (Kim et al., 2019), and parametrized versions of the global optimization benchmark functions (ii) Forrester (1D) (Forrester et al., 2008), (iii) Branin (2D) (Picheny et al., 2013), and (iv) Hartmann-3 (3D) (Szego & Dixon, 1978) as proposed by (Volpp et al., 2020). In Figs. 4a,4b,7, we report the median simple regret, i.e., the difference of the current incumbent value to the function's minimum, over BO iteration. We observe that our GMM-NP model represents a more powerful BO surrogate compared to the baselines, providing further evidence that TRNG-VI with GMM TP approximations yields superior epistemic uncertainty estimates. We provide further results in App. A.5.2, Figs. 7, 8.

## 5.4 DYNAMICS MODELING

We further investigate a challenging dynamics modeling problem on a function class obtained by simulating a Furuta pendulum (Furuta et al., 1992), a highly non-linear 4D dynamical system, as

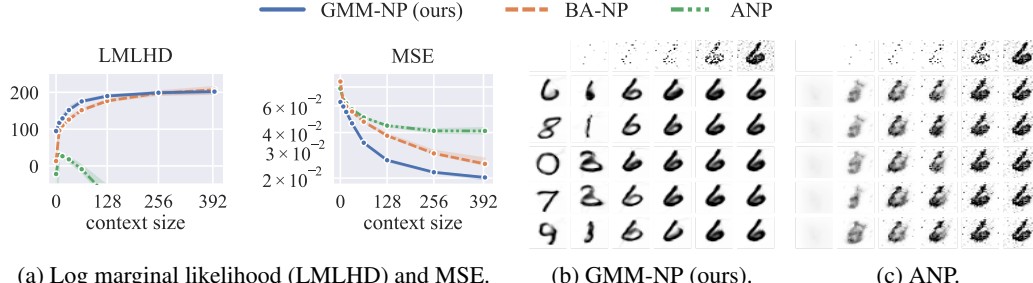

(a) Log marginal likelihood (LMLHD) and MSE.     (b) GMM-NP (ours).     (c) ANP.

Figure 5: Results on 2D image completion on MNIST. Panels (b),(c) visualize predictions on an unseen task showing the digit "6". The first row shows the context pixels, the remaining rows show five corresponding samples. The results are consistent with earlier observations (e.g., Fig. 2): our GMM-NP model shows highly variable samples for small context sets, yielding an accurate estimate of epistemic uncertainty, and contracts properly around the ground truth when more context information is available. ANP yields crisp predictions but massively overfits to the noise, explaining bad LMLHD and MSE scores. We provide further results in App. A.5.3, Fig. 9.

proposed by Volpp et al. (2021). The task is to predict the difference of the next system state $x_{\text{next}} \in \mathbb{R}^4$ to the current system state $x \in \mathbb{R}^4$, i.e., we study one-step ahead dynamics predictions $x \to y = \Delta x \equiv x_{\text{next}} - x \in \mathbb{R}^4$. The function class is generated by simulating $L = 64$ episodes of $N = 64$ timesteps each ($\Delta t = 0.1\,\text{s}$), where for each episode we randomly sample the 7 physical parameters of the pendulum (3 lengths, 2 masses, 2 friction coefficients). The results (Fig. 4c) show that GMM-NP outperforms the baselines in terms of LMLHD by a large margin, demonstrating its applicability to complex dynamics prediction tasks where reliable uncertainty estimates are required, e.g., in robotics applications (Deisenroth et al., 2011). Interestingly, while neither ANP nor BNP can reliable solve this task, BANP performs strongly, reaching GMM-NPs asymptotic performance in terms of LMLHD and yielding even slightly better MSE for small context sets.

## 5.5 IMAGE COMPLETION

To show that our architecture scales to large meta-datasets, we provide results on a 2D image completion experiment on the MNIST database of handwritten digits (LeCun & Cortes, 2010), as proposed by Garnelo et al. (2018b). The task is to predict pixel intensities $y \in \mathbb{R}$ at 2D pixel locations $x \in \mathbb{R}^2$, given a set of context pixels. To obtain a realistic regression task, we add Gaussian noise to each context pixel. The meta-dataset consists of $L = 60000$ images with $N = 784$ pixels each. The results (Fig. 5) are consistent with our previous findings: GMM-NP yields markedly improved performance, outperforming the baselines over the whole range of context sizes. The architectures with deterministic paths ((B)ANP) fail at properly estimating epistemic uncertainties, leading to low LMLHD values, i.p., for large context sizes. Figs. 5b,5c,9 explain why this is the case: GMM-NP (and also, to some extent, BA-NP) generate meaningful images of high variability, corresponding to well-calibrated uncertainty estimates. In contrast, (B)ANP produce essentially deterministic samples that overfit the noise in the context data. While these samples might appear less blurry than those of GMM-NP and BA-NP, they represent inferior solutions of the regression problem.

## 6 CONCLUSION AND OUTLOOK

We proposed GMM-NP, a novel BML algorithm inspired by the NP model architecture. Our approach focuses on accurate task posterior inference, a central algorithmic building block that until now has been treated by amortized inference with set encoders optimized using standard, reparametrized gradients. We demonstrate that this approach leads to suboptimal task posterior approximations and, thus, inefficient optimization of model parameters. We apply modern TRNG-VI techniques that enable expressive variational GMMs, which yields tight ELBOs, efficient optimization, and markedly improved predictive performance in terms of both epistemic uncertainty estimation and accuracy. Despite its simplicity, GMM-NP outperforms the state-of-the-art on a range of experiments and demonstrates its applicability in practical settings, i.p., when meta and context data is scarce. This demonstrates that complex architectural extensions, like Bayesian set encoders or deterministic, attentive computation paths are not required – in fact, we observe that deterministic modules degrade epistemic uncertainty estimation. Therefore, we hope that our work inspires further research on accurate task posterior inference as this turns out to suffice for accurate BML.

REPRODUCIBILITY STATEMENT

We took great care to present a fair comparison of our GMM-NP algorithm with the baseline models, with statistically reliable results that can be easily reproduced. In particular, we

- clearly state the hyperparameter settings and hyperparameter optimization procedure we used (Sec. A.3.1),
- clearly state the generating process for the datasets on which we evaluated our algorithm (Sec. A.4),
- concisely define the evaluation metrics we reported (Sec. A.3.3),
- made sure to evaluate these metrics on large test and sample sets, as well as on multiple (8) random seeds, s.t., our results carry statistical significance (Sec. A.3),
- use source code from the original authors for all baselines (Sec. A.2),
- make the source code for our proposed algorithm available online (Sec. A.2).

ETHICS STATEMENT

We do not expect any negative ethical or societal impact of our work. I.p., we did not use sensitive/personal data in our experiments.

ACKNOWLEDGMENTS

This work was performed on the HoreKa supercomputer funded by the Ministry of Science, Research and the Arts Baden-Württemberg and by the Federal Ministry of Education and Research. The authors further acknowledge support by the state of Baden-Württemberg through bwHPC.

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

# A  APPENDIX

This appendix provides further details that supplement the main part of our paper.

## A.1  ALGORITHMIC DETAILS

In this section we lay out the full set of variational update equations and provide pseudocode for our GMM-NP algorithm.

### A.1.1  VARIATIONAL UPDATE EQUATIONS

We provide the full set of equations required to compute the TRNG update for the variational parameters $\phi_\ell \equiv \{w_{\ell,k}, \boldsymbol{\mu}_{\ell,k}, \boldsymbol{\Sigma}_{\ell,k}\}$, $k \in \{1, \ldots, K\}$, parametrizing our GMM TP approximation as

$$q_{\phi_\ell}(\boldsymbol{z}_\ell) \equiv \sum_k w_{\ell,k} q_{\phi_\ell}(\boldsymbol{z}_\ell | k) \equiv \sum_k w_{\ell,k} \mathcal{N}(\boldsymbol{z}_\ell | \boldsymbol{\mu}_{\ell,k}, \boldsymbol{\Sigma}_{\ell,k}), \quad \sum_k w_{\ell,k} = 1. \tag{10}$$

The TRNG-VI update equations, as proposed by Arenz et al. (2022), read

$$\boldsymbol{\Sigma}_{\ell,k,\text{new}} = \left[ \frac{\eta}{\eta + 1} \boldsymbol{\Sigma}_{\ell,k,\text{old}}^{-1} - \frac{1}{\eta + 1} \boldsymbol{R}_{\ell,k} \right]^{-1}, \tag{11a}$$

$$\boldsymbol{\mu}_{\ell,k,\text{new}} = \boldsymbol{\Sigma}_{\ell,k,\text{new}} \left[ \frac{\eta}{\eta + 1} \boldsymbol{\Sigma}_{\ell,k,\text{old}}^{-1} \boldsymbol{\mu}_{\ell,k,\text{old}} + \frac{1}{\eta + 1} \big( \boldsymbol{r}_{\ell,k} - \boldsymbol{R}_{\ell,k} \boldsymbol{\mu}_{\ell,k,\text{old}} \big) \right], \tag{11b}$$

$$w_{\ell,k,\text{new}} \propto \exp \rho_{\ell,k}, \tag{11c}$$

where $\boldsymbol{R}_{\ell,k}$, $\boldsymbol{r}_{\ell,k}$, and $\rho_{\ell,k}$ are defined as expectations that can be approximated from per-component samples using MC:

$$\boldsymbol{R}_{\ell,k} = \mathbb{E}_{q_{\phi_{\ell,\text{old}}}(\boldsymbol{z}_\ell | k)} \left[ \boldsymbol{\Sigma}_{\ell,k,\text{old}}^{-1} \big( \boldsymbol{z}_\ell - \boldsymbol{\mu}_{\ell,k,\text{old}} \big) \nabla_{\boldsymbol{z}_\ell}^T h_{\ell,k}(\boldsymbol{z}_\ell) \right], \tag{12a}$$

$$\boldsymbol{r}_{\ell,k} = \mathbb{E}_{q_{\phi_{\ell,\text{old}}}(\boldsymbol{z}_\ell | k)} \left[ \nabla_{\boldsymbol{z}_\ell} h_{\ell,k}(\boldsymbol{z}_\ell) \right], \tag{12b}$$

$$\rho_{\ell,k} = \mathbb{E}_{q_{\phi_{\ell,\text{old}}}(\boldsymbol{z}_\ell | k)} \left[ h_{\ell,k}(\boldsymbol{z}_\ell) - \log q_{\phi_{\ell,\text{old}}}(\boldsymbol{z}_\ell | k) \right]. \tag{12c}$$

Here, we defined

$$h_{\ell,k}(\boldsymbol{z}_\ell) \equiv \log \tilde{p}_\ell(\boldsymbol{z}_\ell) + \log q_{\phi_{\ell,\text{old}}}(\boldsymbol{z}_\ell | k) - \log q_{\phi_{\ell,\text{old}}}(\boldsymbol{z}_\ell). \tag{13}$$

The optimal value for the Lagrangean parameter $\eta \geq 0$ that enforces the trust region constraint

$$\text{KL}\left[ q_\phi \| q_{\phi_{\text{old}}} \right] \leq \varepsilon, \tag{14}$$

is defined by a scalar convex optimization problem that can be solved efficiently by a bracketing search, which also ensures positive definiteness of the new covariance matrix $\boldsymbol{\Sigma}_{\ell,k,\text{new}}$.

### A.1.2  GMM INITIALIZATION

We provide details on the initialization of the variational GMMs before meta-training and testing. As we use the same procedure for each task, we drop task indices $\ell$ to avoid clutter. Given a number $K$ of components for the GMM task posterior (TP) $q_\phi(\boldsymbol{z})$ Eq. (7), we use a prior $p(\boldsymbol{z})$ with $K$ components. To initialize the means $\boldsymbol{\mu}_k$, covariances $\boldsymbol{\Sigma}_k$, and mixture weights $w_k$ for $k \in \{1, \ldots, K\}$, we use the same simple heuristic as Arenz et al. (2022):

- Draw the means $\boldsymbol{\mu}_k$ from a $d_z$-dimensional standard Normal distribution,
- The covariances $\boldsymbol{\Sigma}_k$ are initialized as diagonal matrices (with 1 on the diagonal),
- The weights are initialized uniformly as $w_k = 1/K$.

### A.1.3  ALGORITHM SUMMARY

We provide pseudocode for the meta-training stage of our GMM-NP algorithm in Alg. 1 and for the prediction stage in Alg. 2.

---

**Algorithm 1** GMM-NP (Meta-Training)

---

**Require:** Meta-data $\mathcal{D}_\ell = \{\boldsymbol{x}_{\ell,1:N}, \boldsymbol{y}_{\ell,1:N}\}, \ell \in 1 : L$

Sample variably-sized auxiliary tasks $\tilde{\mathcal{D}}_{\tilde{\ell}} = \left\{\boldsymbol{x}_{\tilde{\ell},1:N_{\tilde{\ell}}}, \boldsymbol{y}_{\tilde{\ell},1:N_{\tilde{\ell}}}\right\}, \tilde{\ell} \in 1 : \tilde{L}$, cf. Sec. A.3.2

Initialize variational parameters $\boldsymbol{\phi}_{1:\tilde{L}} = \left\{w_{1:\tilde{L},1:K}, \boldsymbol{\mu}_{1:\tilde{L},1:K}, \boldsymbol{\Sigma}_{1:\tilde{L},1:K}\right\}$

Initialize model parameters $\boldsymbol{\theta}$

**while** not converged **do**

    **for each** minibatch of tasks $I \subset \left\{1, \ldots, \tilde{L}\right\}$ **do**

        Sample $\boldsymbol{z}_{\ell,k,s} \sim q_{\boldsymbol{\phi}_\ell}(\boldsymbol{z}_\ell|k)$ for $\ell \in I, k \in 1 : K, s \in 1 : S$

        Evaluate $h_{\ell,k}$ on $\boldsymbol{z}_{\ell,k,s}$ and $\tilde{\mathcal{D}}_\ell$ for $\ell \in I, k \in 1 : K, s \in 1 : S$, Eq. (13)

        Update variational parameters $\boldsymbol{\phi}_\ell$ for $\ell \in I$, Eq. (8)

        Sample $\boldsymbol{z}_{\ell,s} \sim q_{\boldsymbol{\phi}_\ell}(\boldsymbol{z}_\ell)$ for $\ell \in I, s \in 1 : S$

        Estimate gradient of ELBO Eq. (9): $\nabla_{\boldsymbol{\theta}}\mathcal{L}(\boldsymbol{\theta}) \propto \sum_{s,n} \nabla_{\boldsymbol{\theta}} \log p_{\boldsymbol{\theta}}(\boldsymbol{y}_{\ell,n}|\boldsymbol{x}_{\ell,n}, \boldsymbol{z}_{\ell,s})$

        Perform step in $\boldsymbol{\theta}$ using Adam

    **end for**

**end while**

**return** Model parameters $\boldsymbol{\theta}$

---

**Algorithm 2** GMM-NP (Prediction)

---

**Require:** Context data $\mathcal{D}_*^c = \left\{\boldsymbol{x}_{*,1:M_*}^c, \boldsymbol{y}_{*,1:M_*}^c\right\}$, model parameters $\boldsymbol{\theta}$, target inputs $\boldsymbol{x}_{*,1:N_*}^t$

Initialize variational parameters $\boldsymbol{\phi}_* = \{w_{*,1:K}, \boldsymbol{\mu}_{*,1:K}, \boldsymbol{\Sigma}_{*,1:K}\}$

**while** not converged **do**

    Sample $\boldsymbol{z}_{*,k,s} \sim q_{\boldsymbol{\phi}_*}(\boldsymbol{z}_*|k)$ for $k \in 1 : K, s \in 1 : S$

    Evaluate $h_{*,k}$ on $\boldsymbol{z}_{*,k,s}$ and $\mathcal{D}_*^c$ for $k \in 1 : K, s \in 1 : S$, Eq. (13)

    Update variational parameters $\boldsymbol{\phi}_*$, Eq. (8)

**end while**

Sample $\boldsymbol{z}_* \sim q_{\boldsymbol{\phi}_*}(\boldsymbol{z}_*)$

**return** Predictions $\boldsymbol{y}_{*,n}^t = \mathrm{dec}_{\boldsymbol{\theta}}(\boldsymbol{x}_{*,n}^t, \boldsymbol{z}_*), n \in 1 : N_*$

---

### A.1.4 DISCUSSION OF CONVERGENCE PROPERTIES

**Convergence of the ELBO.** Our algorithm inherits the convergence guarantee of the variational Bayes algorithm as discussed, e.g., in Bishop (2006). In general, convergence of variational Bayes is independent of the concrete optimization strategy used for $(\phi, \theta)$: as long as both the E-step (step in $\phi$) and the M-step (step in $\theta$) increase the ELBO objective (first term in Eq. (9)), the algorithm is guaranteed to converge to a local optimum of the ELBO. While in standard, reparametrized, variational Bayes (as employed by the baseline methods studied in Sec. 5) $(\phi, \theta)$ are optimized jointly using, e.g., Adam (Kingma & Ba, 2015), our method alternates between a step in $\phi$ using TRNG-VI (Arenz et al., 2022) and a step in $\theta$ using Adam. Nevertheless, both steps increase the ELBO, so our algorithm will converge.

**Convergence of the Marginal Likelihood.** As discussed in Sec. 4, our GMM-NP algorithm is designed to improve the convergence behaviour w.r.t. the *marginal likelihood* Eq. (9) in comparison to existing NP-based BML approaches. Recall that the convergence guarantee of the classical expectation-maximization (EM) algorithm w.r.t. the marginal likelihood is lost as soon as the E-step becomes intractable, i.e., as soon as the posterior distribution cannot be computed exactly, and, thus, has to be approximated by a variational distribution, cf., e.g., Bishop (2006). This is the case for most models of reasonable complexity, e.g., for the variational autoencoder (Kingma & Welling, 2013) or the NP model family (Garnelo et al., 2018b). Our GMM-NP model is no exception here, as we build on the NP model for which the TP distribution cannot be computed analytically. Convergence of the marginal likelihood when using the ELBO (first term in Eq. (9)) as a surrogate objective is guaranteed if the ELBO is tight after the E-step, which is the setting of the aforementioned EM algorithm and only the case for a perfect TP approximation, i.e., if $\mathrm{KL}(q_\phi(z)\|p_\theta(z|\mathcal{D}^c)) = 0$, cf. also App. A.3.3. For imperfect approximations, the tightness of the bound is controlled by the variational gap $\mathrm{KL}(q_\phi(z)\|p_\theta(z|\mathcal{D}^c)) > 0$. A better approximate posterior $q_\phi(z)$ yields a tighter ELBO, which in turn brings us closer to the EM setting, i.e., typically improves convergence. Our GMM-NP algorithm builds exactly on this insight: we use an expressive TP approximation by a full-covariance GMM and a powerful optimizer for $\phi$ (TRNG-VI, (Arenz et al., 2022)) to obtain a tighter ELBO than existing BML approaches in order to achieve optimization of the model parameters in a way that efficiently maximizes the marginal likelihood.

## A.2   Baseline Algorithms

Tab. 1 gives an overview of the architectural differences of the BML approaches we compared in our empirical evaluation (Sec. 5).

Table 1: Comparison of state-of-the-art approaches for Bayesian meta-learning (TRNGD = trust region natural gradient descent, RSGD = reparametrized stochastic gradient descent, SGD = stochastic gradient descent, SE = set encoder, MA = mean aggregation, BA = Bayesian aggregation, SA = self attention, CA = cross attention).

|  | TP Approx. | VI Approach | Amortization | Det. Path |
|---|---|---|---|---|
| GMM-NP (ours) | Full-cov. GMM | TRNGD | none | none |
| MA-NP (Garnelo et al., 2018b) | Diag. Gaussian | RSGD | SE + MA | none |
| BA-NP (Volpp et al., 2021) | Diag. Gaussian | RSGD | SE + BA | none |
| BNP (Lee et al., 2020) | Non-parametric | SGD | SE + MA | none |
| ANP (Kim et al., 2019) | Diag. Gaussian | RSGD | SA + SE + MA | CA |
| BANP (Lee et al., 2020) | Non-parametric | SGD | SA + SE + MA | CA |

To compute our results, we consistently use code by the original authors. We also provide source code for our proposed GMM-NP algorithm:

- Source code four our GMM-NP algorithm:
  `https://github.com/ALRhub/gmm_np`
- MA-NP, ANP:
  `https://github.com/deepmind/neural-processes`,
- BA-NP:
  `https://github.com/boschresearch/bayesian-context-aggregation`,
- BNP, BANP:
  `https://github.com/juho-lee/bnp`.

## A.3    EXPERIMENTAL PROTOCOL

To foster reproducibility, we provide details on our experimental protocol.

### A.3.1    MODEL HYPERPARAMETERS

To arrive at a fair comparison of our GMM-NP model with the baseline approaches, we optimize model hyperparameters individually for each model-dataset combination presented in Sec. 5. Concretely, we perform a Bayesian hyperparameter sweep with 256 trials for each model-dataset combination over the parameters detailed below. For the image completion experiment on MNIST, we employ a grid search with fewer trials to keep the computational effort manageable. For hyperparameters not mentioned below, we consistently use standard settings proposed by the original authors. To implement the hyperparameter search, we use the wandb sweep functionality (Biewald, 2020).

**Observation Noise Parametrization.**    As detailed in Sec. 3.2, all compared models (including our GMM-NP) employ a Gaussian likelihood of the form

$$p_{\boldsymbol{\theta}}\left(\boldsymbol{y}|\boldsymbol{x},\boldsymbol{z}\right) \equiv \mathcal{N}\left(\boldsymbol{y}|\text{dec}_{\boldsymbol{\theta}}^{\boldsymbol{\mu}}\left(\boldsymbol{x},\boldsymbol{z}\right),\text{diag}\left(\sigma_{\text{n}}^2\right)\right), \tag{15}$$

where the mean is computed by a decoder DNN $\text{dec}_{\boldsymbol{\theta}}^{\boldsymbol{\mu}}$ receiving the input location $\boldsymbol{x}$ and a latent sample $\boldsymbol{z}$. However, different parametrizations of the observation noise variance $\sigma_{\text{n}}^2$ are used in the literature. As it is not clear which setting is fairest, we also treat the observation noise parametrization as a hyperparameter. Concretely, for each model-dataset combination, we test the following settings for the observation noise (with individual hyperparameter sweeps) and report the best performing one:

1. $\sigma_{\text{n}}^2 = \sigma_{\text{n,true}}^2$ with $\sigma_{\text{n,true}}^2$ being the true noise variance on the data,

2. $\sigma_{\text{n}}^2 \in \mathbb{R}$ is a single float value, optimized jointly with $\boldsymbol{\theta}$,

3. $\sigma_{\text{n}}^2 = \text{dec}_{\boldsymbol{\theta}}^{\boldsymbol{\sigma}}\left(\boldsymbol{x}\right)$, i.e., observation noise is parametrized by a second decoder network, optimized jointly with $\text{dec}_{\boldsymbol{\theta}}^{\boldsymbol{\mu}}$, but receiving only the input location,

4. $\sigma_{\text{n}}^2 = \text{dec}_{\boldsymbol{\theta}}^{\boldsymbol{\sigma}}\left(\boldsymbol{x},\boldsymbol{z}\right)$, i.e., observation noise is parametrized by a second decoder network, optimized jointly with $\text{dec}_{\boldsymbol{\theta}}^{\boldsymbol{\mu}}$, and also receiving both the input location and the latent sample.

For all compared models, and regardless of the parametrization, we bound the observation noise from below using a softplus transformation s.t. $\sigma_{\text{n}} \geq \sigma_{\text{n,min}} = 0.1$, as proposed by (Garnelo et al., 2018b; Kim et al., 2019; Lee et al., 2020).

**DNN Architectures.**    For all experiments and all baseline models, we use encoder and decoder DNNs with two hidden layers. Likewise, our GMM-NP model uses a decoder DNN with two hidden layers. We optimize the number of hidden units per layer within the bounds $\{8, \ldots, 64\}$.

**Latent Dimensionalities.**    For baseline models with parametric latent distributions (all except B(A)NP), we optimize the latent dimension $d_z$ within the bounds $\{1, \ldots, 64\}$. As our GMM-NP algorithm employs full covariance matrices, we restrict the bounds for $d_z$ to $\{1, \ldots, 8\}$ for a fair comparison.

**Number of GMM components.**    For our GMM-NP algorithm, as well as for iBayes-GMM (Lin et al., 2020) used for the comparison in Sec. A.5.1, we optimize the number of GMM components within the bounds $\{1, \ldots, 10\}$.

**Learning Rates and Trust Region Bounds.**    All algorithms use the Adam optimizer with standard settings to update DNN weights. We optimize the corresponding learning rates on a log-uniform scale within the bounds $\left[10^{-5}, 10^{-1}\right]$. We use the same settings to optimize the step size for the GMM updates of the variational parameters of the iBayes-GMM algorithm (Lin et al., 2020) used for the comparison in Sec. A.5.1. As proposed by Arenz et al. (2022), we optimize the Lagrangean parameter $\eta$ of our GMM-NP algorithm using a bracketing search on the interval $\left[10^{-3}, 10^{-1}\right]$.

A.3.2    Auxiliary Subtask Generation for Meta-Training

We describe the procedure to sample auxiliary subtasks during meta-training in more detail, cf. Sec. 4.

**Nomenclature.** Recall from Sec. 3.2 that we define a *meta-task* as the set of all available (noisy) evaluations $\mathcal{D}_\ell$, $\ell \in \{1, \ldots, L\}$ from an unknown function $f_\ell$ and that each meta-task contains $N$ examples. Thus, a meta-task $\mathcal{D}_\ell$ is all data a BML algorithm has available to learn about $f_\ell$ during meta-training. A *subtask* of meta-task $\mathcal{D}_\ell$ is defined as an arbitrary subset of $\mathcal{D}_\ell$.

**Auxiliary Subtask Sampling.** As described in Sec. 4, standard NP meta-training samples auxiliary subtasks from the metadata for each minibatch step in order to provide the decoder with samples from task posterior approximations informed by a range of context sizes. We use the following standard procedure (Garnelo et al., 2018b; Kim et al., 2019; Lee et al., 2020) to sample auxiliary subtasks to evaluate the optimization objectives of the baseline approaches (e.g., Eq. 5 for standard NP). Given a minibatch $I \subset \{1, \ldots, L\}$ of meta-tasks $\mathcal{D}_\ell$, $\ell \in I$, we first sample auxiliary subtasks $\tilde{\mathcal{D}}_\ell$ with a size $\tilde{N}$ drawn uniformly from $\tilde{N} \in \{N_{\min} + 1, \ldots, N_{\max}\}$ with $N_{\min} \geq 1$ and $N_{\max} \leq N$. Then, we sample context sets $\tilde{\mathcal{D}}_\ell^c \subset \tilde{\mathcal{D}}_\ell$ of size $M$, drawn uniformly from $M \in \left\{ 1, \ldots, \tilde{N} \right\}$. $\tilde{\mathcal{D}}_\ell^c$ and $\tilde{\mathcal{D}}_\ell$ are then used in Eq. 5 to compute the ELBO objective for the current minibatch.

As described in Sec. 4, our GMM-NP algorithm uses a similar approach: we employ auxiliary subtasks with sizes $\tilde{N}$ drawn uniformly from $\tilde{N} \in \{N_{\min}, \ldots, N_{\max}\}$ to evaluate the updates for the variational GMM parameters and the model parameters. Note that our algorithm does not require to sample context sets during meta training from the auxiliary subtasks. Furthermore, recall that we train one variational GMM for each auxiliary subtask and retain those GMMs over the whole course of meta training, so we fix a set of $\tilde{L}$ auxiliary subtasks at the beginning of meta-training (in contrast to standard NPs, which sample new subtasks for each minibatch).

We use the following settings for $N_{\min}$, $N_{\max}$ in our experiments: $N_{\min} = 1$, $N_{\max} = N$, except for MNIST image completion where we use $N_{\max} = N/2$.

Further, we use $\tilde{L} = 32L$, except for MNIST image completion where we use $\tilde{L} = 8$.

A.3.3    Metrics

For each model-dataset combination, we retrain the best hyperparameter setting determined according to Sec. A.3.1 with $8$ different random seeds used for model initialization, and report the median value together with $(5\%, 95\%)$ percentiles of the metrics computed according to the formulae provided below. For all experiments (except the MNIST image completion experiment), we evaluate all metrics on $L = 256$ unseen test tasks $\mathcal{D}_{1:L}$ with $\mathcal{D}_\ell = \{\boldsymbol{y}_{\ell,1:N}, \boldsymbol{x}_{\ell,1:N}\}$ and $N = 64$, from which we sample context sets $\mathcal{D}_\ell^c \subset \mathcal{D}_\ell$. For the image completion experiment we use $L = 1024$ and $N = 784$ (the number of pixels per image). We report the results in dependence of the context set size.

**Log Marginal Predictive Likelihood (LMLHD).** For a given task $\ell$ the LMLHD is defined by Eq. (4), which we restate here for convenience:

$$\log q_{\boldsymbol{\theta}} \left( \boldsymbol{y}_{\ell,1:N} | \boldsymbol{x}_{\ell,1:N}, \mathcal{D}_\ell^c \right) \equiv \log \int \prod_n p_{\boldsymbol{\theta}} \left( \boldsymbol{y}_{\ell,n} | \boldsymbol{x}_{\ell,n}, \boldsymbol{z}_\ell \right) q \left( \boldsymbol{z}_\ell | \mathcal{D}_\ell^c \right) \mathrm{d}\boldsymbol{z}_\ell. \tag{16}$$

Here, we use the generic notation $q \left( \boldsymbol{z}_\ell | \mathcal{D}_\ell^c \right)$ to denote the task posterior TP approximation, the concrete form of which depends on the BML model under consideration. As the integral is analytically intractable, we resort to an MC approximation. To this end, we sample $S = 1024$ samples

$\boldsymbol{z}_{\ell,s} \sim q\left(\boldsymbol{z}_\ell | \mathcal{D}_\ell^c\right)$ in the test set and compute (Volpp et al., 2021)

$$\log q_{\boldsymbol{\theta}}\left(\boldsymbol{y}_{\ell,1:N} | \boldsymbol{x}_{\ell,1:N}, \mathcal{D}_\ell^c\right) \equiv \log \int \prod_{n=1}^{N} p_{\boldsymbol{\theta}}\left(\boldsymbol{y}_{\ell,n} | \boldsymbol{x}_{\ell,n}, \boldsymbol{z}_\ell\right) q\left(\boldsymbol{z}_\ell | \mathcal{D}_\ell^c\right) \mathrm{d}\boldsymbol{z}_\ell \tag{17}$$

$$\approx \log \frac{1}{S} \sum_{s=1}^{S} \prod_{n=1}^{N} p_{\boldsymbol{\theta}}\left(\boldsymbol{y}_{\ell,n} | \boldsymbol{x}_{\ell,n}, \boldsymbol{z}_{\ell,s}\right) \tag{18}$$

$$= -\log S + \operatorname*{logsumexp}_{s=1}^{S} \sum_{n=1}^{N} \log p_{\boldsymbol{\theta}}\left(\boldsymbol{y}_{\ell,n} | \boldsymbol{x}_{\ell,n}, \boldsymbol{z}_{\ell,s}\right). \tag{19}$$

where logsumexp denotes the numerically stable implementation of the function $\log \sum_s \exp(x_s)$, available in any scientific computing framework. We then compute the median of this expression over all tasks of the test set.

**Mean Squared Error (MSE).** We report the MSE w.r.t. the mean prediction. That is, for a given task $\ell$, we again draw $S = 1024$ samples $\boldsymbol{z}_{\ell,s} \sim q\left(\boldsymbol{z}_\ell | \mathcal{D}_\ell^c\right)$ and compute

$$\mathrm{MSE}\left(\boldsymbol{y}_{\ell,1:N}, \boldsymbol{x}_{\ell,1:N}\right) \equiv \frac{1}{N} \sum_{n=1}^{N} \left(\frac{1}{S} \sum_{s=1}^{S} \mathrm{dec}_{\boldsymbol{\theta}}^{\boldsymbol{\mu}}\left(\boldsymbol{x}_{\ell,n}, \boldsymbol{z}_{\ell,s}\right) - \boldsymbol{y}_{\ell,n}\right)^2. \tag{20}$$

We then compute the median of this expression over all tasks of the test set.

**ELBO Looseness.** For a given task $\ell$, we define the ELBO looseness as the KL-divergence between the approximate and true task posteriors. According to Eq. (4), this decomposes as

$$\mathrm{KL}\left[q\left(\boldsymbol{z}_\ell | \mathcal{D}_\ell\right) \| p_{\boldsymbol{\theta}}\left(\boldsymbol{z}_\ell | \mathcal{D}_\ell\right)\right] = \log q_{\boldsymbol{\theta}}\left(\boldsymbol{y}_{\ell,1:N} | \boldsymbol{x}_{\ell,1:N}, \mathcal{D}_\ell^c\right) \tag{21}$$

$$- \mathbb{E}_{q(\boldsymbol{z}_\ell | \mathcal{D}_\ell)}\left[\sum_{n=1}^{N} \log p_{\boldsymbol{\theta}}\left(\boldsymbol{y}_{\ell,n} | \boldsymbol{x}_{\ell,n}, \boldsymbol{z}_\ell\right) + \log \frac{q\left(\boldsymbol{z}_\ell | \mathcal{D}_\ell^c\right)}{q\left(\boldsymbol{z}_\ell | \mathcal{D}_\ell\right)}\right], \tag{22}$$

with $\log q_{\boldsymbol{\theta}}\left(\boldsymbol{y}_{\ell,1:N} | \boldsymbol{x}_{\ell,1:N}, \mathcal{D}_\ell^c\right)$ defined by Eq. (16). The second term is the ELBO,

$$\mathcal{L}\left(\boldsymbol{\theta}, \mathcal{D}_\ell^c, \mathcal{D}_\ell\right) \equiv \mathbb{E}_{q(\boldsymbol{z}_\ell | \mathcal{D}_\ell)}\left[\sum_{n=1}^{N} \log p_{\boldsymbol{\theta}}\left(\boldsymbol{y}_{\ell,n} | \boldsymbol{x}_{\ell,n}, \boldsymbol{z}_\ell\right) + \log \frac{q\left(\boldsymbol{z}_\ell | \mathcal{D}_\ell^c\right)}{q\left(\boldsymbol{z}_\ell | \mathcal{D}_\ell\right)}\right], \tag{23}$$

where we made its dependence on both the test set $\mathcal{D}_\ell$ and the context set $\mathcal{D}_\ell^c \subset \mathcal{D}_\ell$ explicit (in contrast to our notation in the main part of this paper). We say the ELBO is tight if its looseness is zero. Then, $\log q_{\boldsymbol{\theta}}\left(\boldsymbol{y}_{\ell,1:N} | \boldsymbol{x}_{\ell,1:N}, \mathcal{D}_\ell^c\right) = \mathcal{L}\left(\boldsymbol{\theta}, \mathcal{D}_\ell^c, \mathcal{D}_\ell\right)$, and optimization of the ELBO w.r.t. $\boldsymbol{\theta}$ is equivalent to optimization of the LMLHD.

For our ablation study (Sec. 5.2), we estimate the looseness of the ELBO by computing the difference of an importance-weighted MC estimate with proposal distribution $q\left(\boldsymbol{z}_\ell | \mathcal{D}_\ell\right)$ of the LMLHD and an MC estimate of the ELBO Eq. (23) with $S = 1024$ samples $\boldsymbol{z}_{\ell,s} \sim q\left(\boldsymbol{z}_\ell | \mathcal{D}_\ell\right)$.

A.4   DATA GENERATION

We provide details on the meta-datasets we use to train the models we compare in Sec. 5. Concretely, we provide

- the dimension $d_x$ of inputs $\boldsymbol{x}_{\ell,n} \in \mathbb{R}^{d_x}$,
- the domain $\mathcal{C} \subset \mathbb{R}^{d_x}$ from which we uniformly sample $\boldsymbol{x}_{\ell,n}$,
- the dimension $d_y$ of targets $\boldsymbol{y}_{\ell,n} \in \mathbb{R}^{d_y}$,
- an expression for the function $f_\ell : \mathbb{R}^{d_x} \to \mathbb{R}^{d_y}$, s.t., $\boldsymbol{y}_{\ell,n} = f_\ell(\boldsymbol{x}_{\ell,n}) + \boldsymbol{\varepsilon}_n$,
- the noise standard deviation $\sigma$, s.t., $\boldsymbol{\varepsilon}_n \sim \mathcal{N}(\boldsymbol{0}, \sigma^2)$,
- the number $L$ of meta-tasks and the number $N$ of datapoints for each meta-task.

We denote the uniform distribution on $(a, b)^d \subset \mathbb{R}^d$ by $\mathrm{U}(a, b)^d$.

**Sinusoidal Functions.**

- $d_x = 1$
- $\mathcal{C} = [-5.0, 5.0]$
- $d_y = 1$
- $f_\ell(x) = A_\ell \sin(x - \phi_\ell)$, $A_\ell \sim \mathrm{U}(0.1, 5.0)$, $\phi_\ell \sim \mathrm{U}(0.0, \pi)$
- $\sigma = 0.25$
- $L = 64$, $N = 16$

**Mix of Affine and Sinusoidal Functions.**

- $d_x = 1$
- $\mathcal{C} = [-5.0, 5.0]$
- $d_y = 1$
- $f_\ell^1(x) = a_\ell x + b_\ell$, $a_\ell \sim \mathrm{U}(-3.0, 3.0)$, $b_\ell \sim \mathrm{U}(-3.0, 3.0)$,
  $f_\ell^2(x) = A_\ell \sin(x - \phi_\ell)$, $A_\ell \sim \mathrm{U}(0.1, 5.0)$, $\phi_\ell \sim \mathrm{U}(0.0, \pi)$
  $f_\ell$ is given either by $f_\ell^1$ or $f_\ell^2$ with probability 0.5.
- $\sigma = 0.25$
- $L = 64$, $N = 16$

**RBF-GP samples.**

- $d_x = 1$
- $\mathcal{C} = [-2.0, 2.0]$
- $d_y = 1$
- $f_\ell$ is drawn from a Gaussian process prior with RBF kernel with lengthscale $l_\ell \sim \mathrm{U}(0.5, 1.0)$ and signal variance $s_\ell \sim \mathrm{U}(0.5, 1.0)$.
- $\sigma = 0.1$
- $L = 64$, $N = 16$

**Forrester 1D.**

- $d_x = 1$
- $d_y = 1$
- We use the parametrized Forrester function Forrester et al. (2008) as defined on
  `https://www.sfu.ca/~ssurjano/forretal08.html`.
- $\sigma = 0.25$
- $L = 64$, $N = 16$

**Branin 2D.**

- $d_x = 2$
- $d_y = 1$
- We use the definition given on `https://www.sfu.ca/˜ssurjano/branin.html` and apply translations $\boldsymbol{\tau}_\ell \sim \mathrm{U}\left(-0.25, 0.25\right)^2$ to $\boldsymbol{x}$, and scale the function values by $s_\ell \sim \mathrm{U}\left(0.75, 1.25\right)$.
- $\sigma = 0.25$
- $L = 64, N = 16$

**Hartmann 3D.**

- $d_x = 3$
- $d_y = 1$
- We use the definition given on `https://www.sfu.ca/˜ssurjano/hart3.html` and apply translations $\boldsymbol{\tau}_\ell \sim \mathrm{U}\left(-0.25, 0.25\right)^3$ to $\boldsymbol{x}$, and scale the function values by $s_\ell \sim \mathrm{U}\left(0.75, 1.25\right)$.
- $\sigma = 0.1$
- $L = 64, N = 16$

**4D Furuta Dynamics Prediction.**

- $d_x = 4$
- $d_y = 4$
- We use the dynamics equations given in Cazzolato & Prime (2011) to simulate episodes, starting from the pendulum balancing in the upright position. The input is the current system state $\boldsymbol{x} \in \mathbb{R}^4$, the target is the difference to the next system state $\boldsymbol{x}_{\mathrm{next}} \in \mathbb{R}^4$, i.e., $\boldsymbol{y} = \Delta\boldsymbol{x} \equiv \boldsymbol{x}_{\mathrm{next}} - \boldsymbol{x} \in \mathbb{R}^4$.
- Noise is generated by random actions on the joints.
- $L = 64, N = 64$

**2D MNIST Image Completion.**

- $d_x = 2$
- $d_y = 1$
- We use the MNIST handwritten image database (LeCun & Cortes, 2010). Each image corresponds to one task. The input $\boldsymbol{x}$ is the pixel location, the target $y$ is the pixel intensity.
- $\sigma = 0.25$
- $L = 60000, N = 784$

## A.5 FURTHER EXPERIMENTAL RESULTS

We provide further experimental results for the experiments presented in Sec. 5.

### A.5.1 ABLATION: TRUST REGIONS

In Fig. 6 we compare two methods for step size control for natural gradient VI, namely direct step size control as proposed by (Lin et al., 2020) and trust region step size control (Arenz et al., 2022), as used by our GMM-NP algorithm. We observe that trust regions lead to more robust optimization of the variational parameters, and, thus, to tighter ELBOs. This allows more efficient optimization of the model parameters, leading to improved predictive performance.

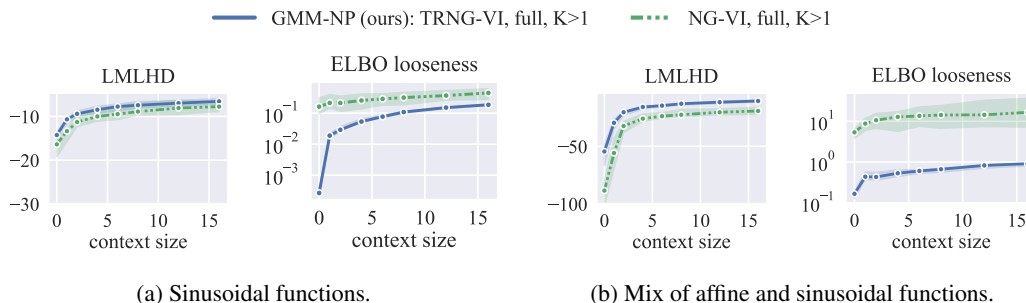

(a) Sinusoidal functions.          (b) Mix of affine and sinusoidal functions.

Figure 6: Log marginal predictive likelihood (LMLHD) and ELBO looseness over context size for our trust region natural gradient VI (TRNG-VI)-based (Arenz et al., 2022) GMM-NP algorithm in comparison to iBayes-GMM (Lin et al., 2020) that uses direct step size control instead of trust regions (NG-VI). Trust regions improve variational optimization, leading to tighter ELBOs, and, consequently, to improved predictive performance.

### A.5.2 BAYESIAN OPTIMIZATION EXPERIMENTS

We provide the full set of results for our Bayesian optimzation experiments, cf. Sec. 5.3: Fig. 7 shows the optimization regret for all four evaluated function classes, and Fig. 8 the corresponding results for LMLHD and MSE.

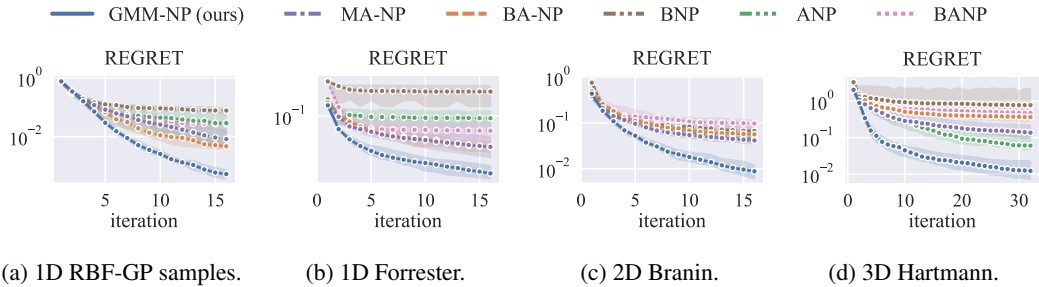

Figure 7: Simple regret over optimization iteration, when using BML models as Bayesian Optimization (BO) surrogates on various function classes. As BO relies on well-calibrated uncertainty predictions, the results demonstrate that GMM-NP provides superior uncertainty estimates.

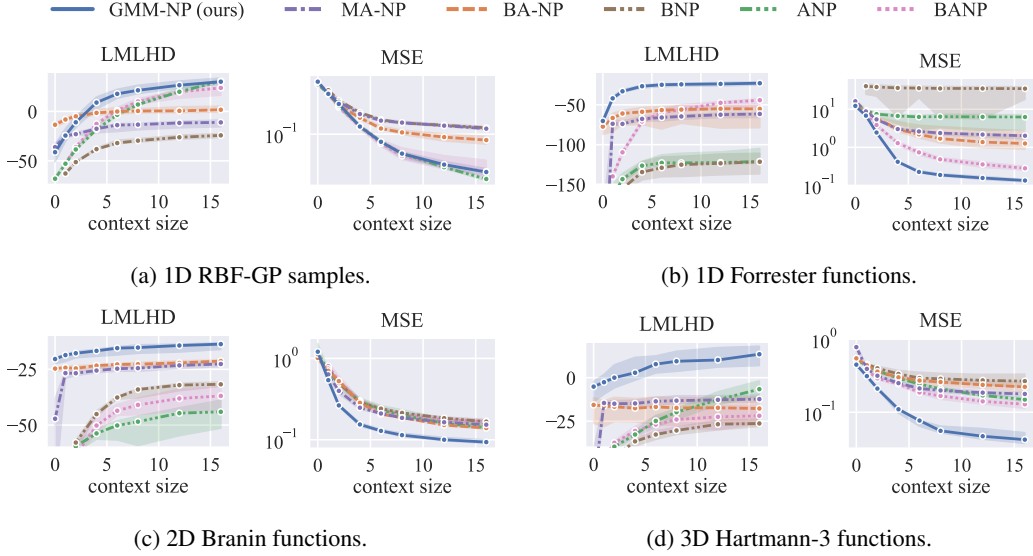

Figure 8: Log marginal predictive likelihood (LMLHD) and mean squared error (MSE) over context size on various function classes. GMM-NP generally performs favorably, showing accurate predictions with well-calibrated uncertainties.

### A.5.3 2D Image Completion on MNIST

Fig. 9 shows the full set of predictions on the 2D image completion experiment on MNIST.

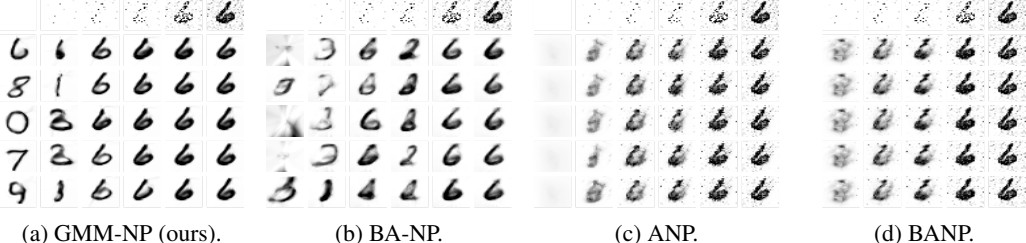

   (a) GMM-NP (ours).          (b) BA-NP.               (c) ANP.              (d) BANP.

Figure 9: Predictions on an unseen instance of the MNIST 2D image completion task, showing the digit "6". The first row of each panel shows the context pixels (ranging from zero pixels in the left column to the full image in the right column). The remaining rows show five samples from the BML models, conditioned on the context pixels shown in the first row. The results are consistent with observations from the other experiments (e.g., Fig. 10): our GMM-NP model shows highly variable samples for small context sets, yielding an accurate estimate of epistemic uncertainty, and contracts properly around the ground truth when more context information is available. BA-NP also shows variable samples, albeit of lower quality. ANP and BANP yield crisp predictions but massively overfit to the noise, explaining their low LMLHD scores. Note also that BANP does not allow predictions for empty context sets.

### A.5.4 VISUALIZATION OF MODEL PREDICTIONS

Figs. 10 and 11 show further visualization of predictions of models trained on the mix of affine and sinusoidal functions.

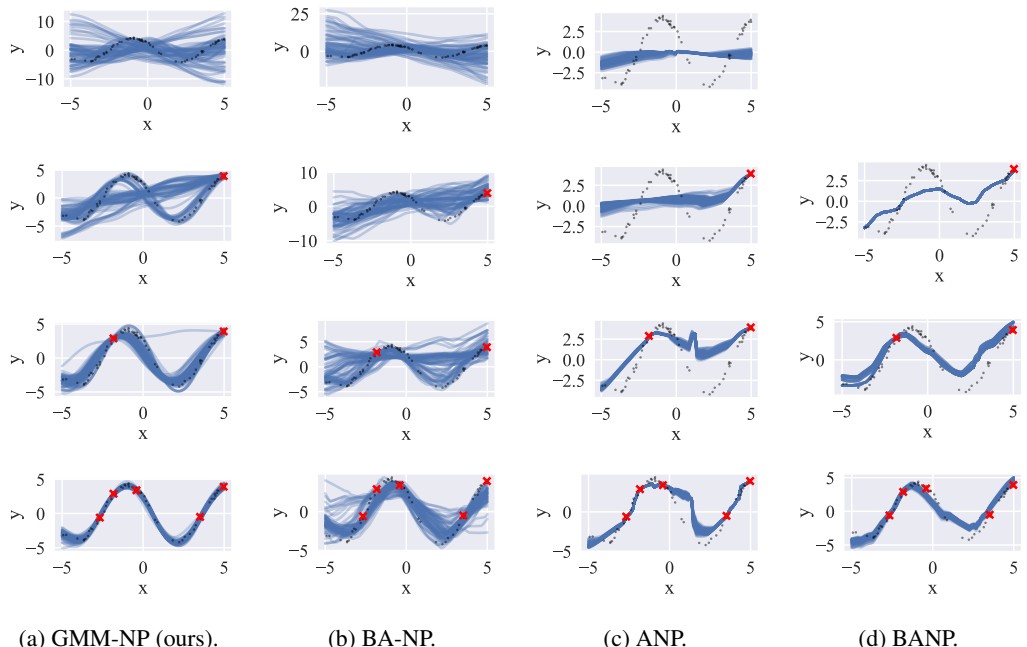

(a) GMM-NP (ours).      (b) BA-NP.      (c) ANP.      (d) BANP.

Figure 10: Function samples computed by various BML models (columns), trained on a function class consisting of a mix of affine and sinusoidal functions (cf. Sec. 5.1), when provided with increasing amounts of context examples (red crosses, rows) from an unseen sinusoidal representative function. We observe that our GMM-NP model accurately quantifies epistemic uncertainty through the variability of its function samples. BA-NP also shows variable samples, but does not achieve the same predictive performance due to its inaccurate approximation of the task posterior distribution. ANP and BANP, both of which employ deterministic computation paths with attention modules, produce essentially deterministic predictions that massively overfit the context data and fail to give a reasonable estimate of the predictive distribution. Therefore, these models have to quantify epistemic uncertainty through the likelihood noise variance, which is ineffective, cf. Fig. 11. Note also that BANP does not provide predictions for empty context sets.

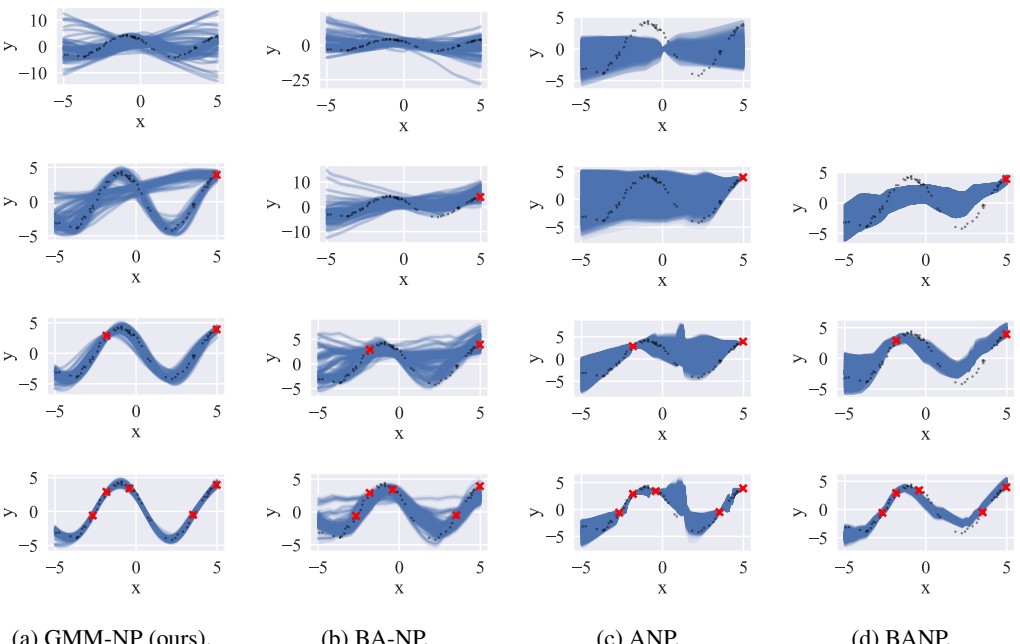

(a) GMM-NP (ours).          (b) BA-NP.          (c) ANP.          (d) BANP.

Figure 11: This figure shows the same data as Fig. 10, but for each function sample we also show a band of $\pm 1$ standard deviation of the observation noise, as computed by the decoder DNN. GMM-NP quantifies epistemic uncertainty correctly through its task posterior approximation, and thus does not have to rely on the decoder DNN to quantify epistemic uncertainty through the observation noise. In contrast, ANP and BANP fail to produce variable function samples, and have to make up for that by quantifying epistemic uncertainty through the observation noise, which is ineffective. Note also that BANP does not provide predictions for empty context sets.

### A.5.5 VISUALIZATION OF LATENT SPACE STRUCTURE

We provide further visualizations similar to Fig. 1 of the task posterior approximation and corresponding function samples of our GMM-NP, when trained on the sinusoidal function class.

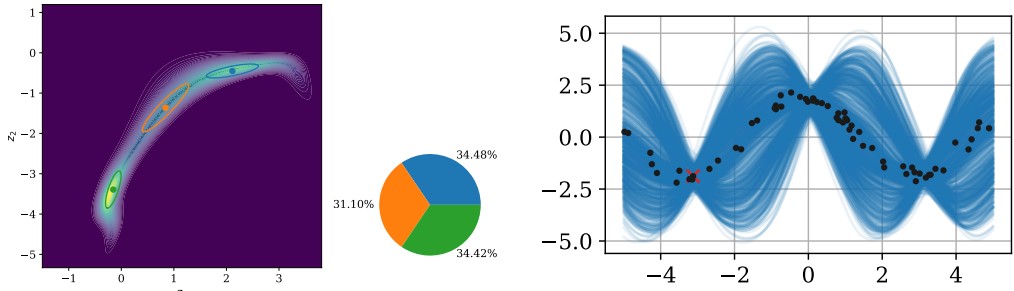

(a) A small context set (one single example indicated by the red cross) yields a highly correlated, multi-modal task posterior distribution. Our GMM approximation correctly captures this, s.t., amplitudes and phases of the predicted sinusoidal functions are in accordance with the observed context data point.

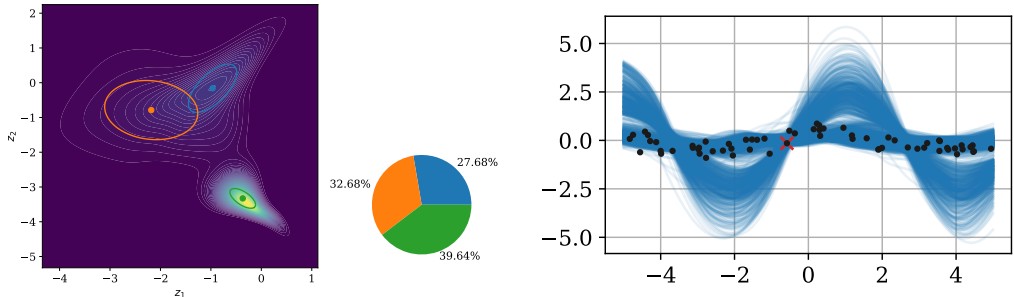

(b) A second example on another instance of the sinusoidal function class, where the task posterior shows pronounced multimodality, which translates into a bimodal predictive distribution.

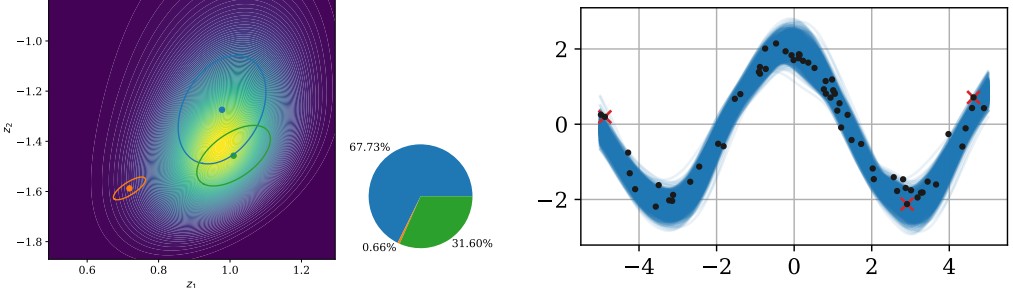

(c) Larger context sizes (three examples, red crosses) leave less task ambiguity, resulting in a unimodal and nearly isotropic task posterior distribution. Our GMM approximation again correctly approximates this distribution, making use of only two of the $K = 3$ mixture components (the mixture weight of the orange component is close to zero, so no samples from this component are observed).

Figure 12: Visualization of our GMM-NP model for a $d_z = 2$ dimensional latent space, trained on sinusoidal functions with varying amplitudes and phases, cf. Sec. 5.1. Left panels: unnormalized task posterior distribution (contours) and variational GMM approximation with $K = 3$ components (ellipses, mixture weights in %). Right panels: corresponding samples from our model (blue lines), when having observed a context data set (red crosses), together with unobserved ground truth data (black dots). The visualizations show that (i) the true task posterior distribution can be highly correlated and multimodal, i.p., for small context sets (panels a,b), (ii) our variational task posterior approximation correctly approximates this distribution, which (iii) leads to expressive predictive distributions that incorporate both the inductive priors learned from the meta-dataset (all samples are sinusoidal in shape) and the additional information contained in the context set (all samples pass close to the context data).

A.5.6 RUNTIME COMPARISON

**Discussion of Limitations.** As the meta-training stage of GMM-NP requires computational effort comparable to standard NP (cf. Sec. 4), the only computational overhead of our algorithm occurs at test time, due to the optimization loop required to fit a variational GMM to $\mathcal{D}_*^c$. While this can be trivially parallelized for multiple test tasks, it incurs a higher computational burden in comparison to the single forward pass through NP's set encoder (we provide an evaluation of the runtime of our algorithm on the synthetic tasks studied in Sec. 5.1 below). We leave a detailed examination for future work, but mention two possible remedies: (i) for problems where test data arrives sequentially, we expect that a few update steps in $\phi_*$ suffice to reach convergence, and (ii) it might be possible to find amortized approximations to Eqs. (8), similar in spirit to standard NP, that retain the advantages of TRNG-VI.

**Meta-Training.** Fig. 13 shows the learning curves for meta-training corresponding to the results presented in the main part of this paper. As discussed in Sec. 4, GMM-NP incurs a computational cost comparable to the baseline methods.

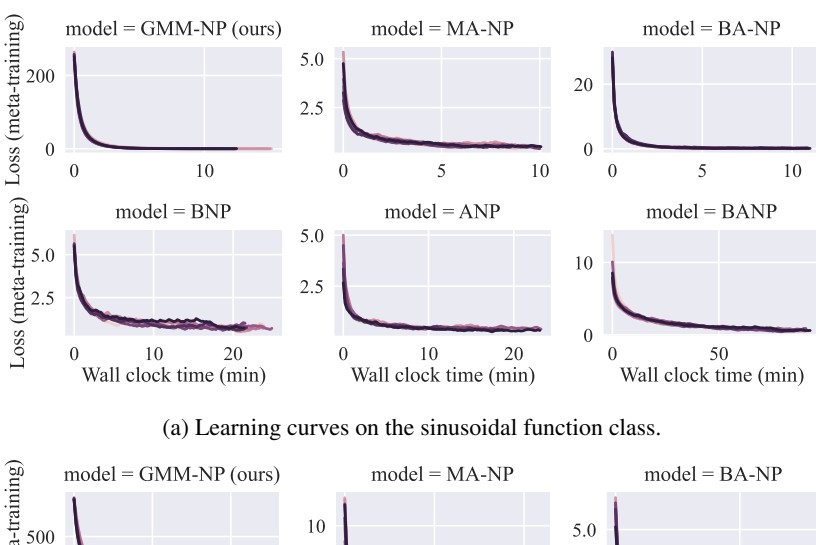

(a) Learning curves on the sinusoidal function class.

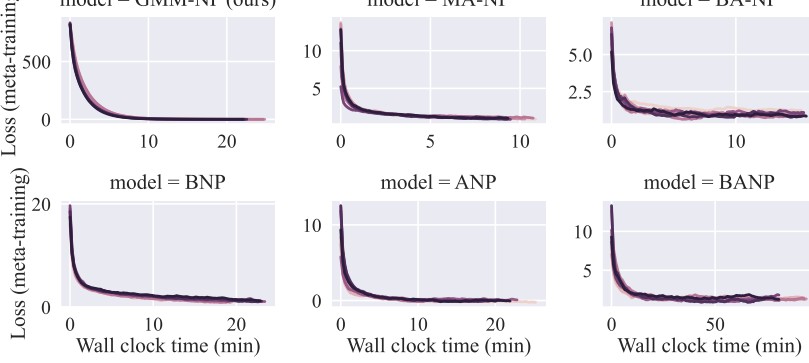

(b) Learning curves on the mix of affine and sinusoidal function class.

Figure 13: Learning curves for meta-training on the synthetic datasets, cf. Sec. 5.1. For each method, we show the learning curves for the 8 seeds used to compute the results presented in the main text. For GMM-NP, we show the loss for the decoder parameters $\theta$, for the other methods we show the joint loss for the encoder and decoder parameters $(\phi, \theta)$. Note that for GMM-NP, convergence of $\theta$ implies convergence of the variational parameters $\phi$. As discussed in Sec. 4, GMM-NP incurs a computational cost comparable to the baseline methods.

**Test-time Adaptation.** As discussed in Sec. 4, GMM-NP does not amortize TP inference, i.e., it does not learn a set encoder architecture, but adapts new variational GMMs at test time. Naturally, this incurs a higher computational cost in comparison to amortized architectures, which compute

predictions on test tasks in a single forward pass through their set-encoder – decoder architecture. In Fig. 14, we show the learning curves for fitting variational GMMs (by iterating Eqs. (8)) to the test tasks and for the range of context sizes used to compute the results presented in Sec. 5.1. GMM-NP's TRNG-VI optimization converges in approx. $0.1\,\mathrm{s} - 1\,\mathrm{s}$ per test task (depending on the context set size).

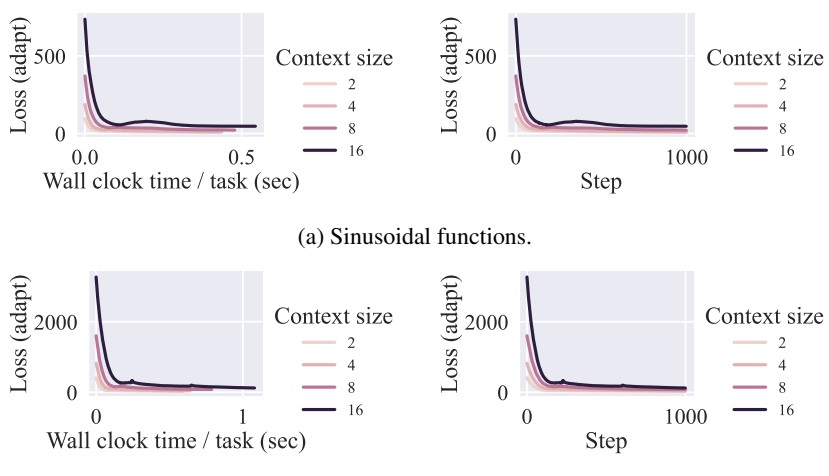

(a) Sinusoidal functions.

(b) Mix of affine and sinusoidal functions.

Figure 14: Learning curves for fitting variational GMMs to the test tasks by TRNG-VI (Arenz et al., 2022), as used by our GMM-NP (Sec. 4), on the synthetic datasets (Sec. 5.1). The quantity labelled "Loss (adapt)" is the expected negative log density of the unnormalized TP under the GMM TP approximation. Note that this is not the loss function optimized by iterating Eqs. (8), but it serves as a proxy to judge convergence. We show results in terms of wall clock time per test task (left panels) and in terms of TRNG-VI steps (right panels), for the range of context sizes used to compute the results in the main text. GMM-NP's TRNG-VI optimization converges in approx. $0.1\,\mathrm{s} - 1\,\mathrm{s}$ per test task (depending on the context set size).

A.5.7  ANALYSIS OF HPO RESULTS

As discussed in Secs. 5 and A.3, we optimized architectural hyperparameters individually for each model-dataset combination presented in our empirical evaluation, in order to arrive at a fair comparison of our GMM-NP with the baseline methods. In Tab. 2, we provide the resulting settings for the latent dimensionality $d_z$, and the number of parameters of the BML models compared in Sec. 5.1. While the number of variational parameters during meta-training is naturally comparably high for non-amortizing methods such as GMM-NP, we observe that the expressive GMM-NP TP approximation allows comparably lightweight decoders and small latent dimensions. This is intuitive, as simple TP approximations require (i) large latent dimensions to encode relevant information in the latent space, together with (ii) expressive decoder architectures to transform the simple latent distribution into an expressive predictive distribution. Note further that the variational parameters belonging to different tasks are not coupled in non-amortizing architectures such as ours, which allows trivial parallelization of the variational optimization between tasks, explaining why the computational cost is easily managable, cf. Sec. A.5.6. Note also that the number of variational parameters one has to store and adapt for GMM-NP to make predictions on unseen test tasks is comparably small because the variational GMMs learned during meta-training can be discarded as they are not required for predictions at test time.

Table 2: Results of our hyperparameter optimization on the sinusoidal function class and on the mix of affine and sinusoidal functions. We provide the settings for the latent dimensionality $d_z$ and the number of parameters of the BML models compared in Sec. 5.1 (i.e., the number of decoder parameters $|\boldsymbol{\theta}|$ as well as the number of encoder / variational parameters $|\boldsymbol{\phi}|$). If attentive modules are present, their parameters are counted as being part of the encoder. For our GMM-NP, we also provide the number of GMM-components $K$. Furthermore, as GMM-NP does not amortize TP-inference but learns separate variational GMMs for each subtask generated from the meta-dataset (cf. Secs. 4 and A.3.2), we also provide the total number of variational GMM parameters during meta-training. Note that these variational GMMs are decoupled and can be optimized in parallel. Furthermore they are not required for predictions at test time and can be discarded after meta-training.

|  |  | $d_z$ | $K$ | $|\boldsymbol{\theta}|$ | $|\boldsymbol{\phi}|$ (per task) | $|\boldsymbol{\phi}|$ (meta-training) |
|---|---|---|---|---|---|---|
| Sinusoid | GMM-NP (ours) | 3 | 4 | 121 | 39 | 79872 |
|  | MA-NP | 10 | - | 2298 | 2595 | 2595 |
|  | BA-NP | 27 | - | 7334 | 7386 | 7386 |
|  | BNP | 56 | - | 12770 | 19488 | 19488 |
|  | ANP | 19 | - | 4096 | 10335 | 10335 |
|  | BANP | 40 | - | 6562 | 28320 | 28320 |
| Line-Sine | GMM-NP (ours) | 4 | 4 | 2289 | 59 | 120832 |
|  | MA-NP | 19 | - | 2562 | 3679 | 3679 |
|  | BA-NP | 30 | - | 11592 | 11650 | 11650 |
|  | BNP | 54 | - | 11882 | 18144 | 18144 |
|  | ANP | 30 | - | 4496 | 14448 | 14448 |
|  | BANP | 32 | - | 4226 | 18304 | 18304 |

