# OpenReview forum: "Accurate Bayesian Meta-Learning by Accurate Task Posterior Inference"
_ICLR.cc/2023/Conference — ICLR 2023 poster_

### Official Review · Reviewer_7Kjw · 2022-10-22

**Confidence:** 4
**Correctness:** 4
**Technical Novelty And Significance:** 3
**Empirical Novelty And Significance:** 3
**Recommendation:** 8

**Clarity, Quality, Novelty And Reproducibility:**

**Clarity**: I can understand the overall ideas of the paper after some careful thinkings. It's a bit painfully structured, trying to explain both the idea itself and its difference with existing works. It could be beneficial to present the work itself first and then have a separate section to compare against existing works.

**Quality**: The paper tests the performance against existing methods extensively, to justify the effectiveness of the different aspects of contributions made.

**Novelty**: The application of TRNG variational inference method for GMM in BML is the main novelty point of this work. Although it does not sound super groundbreaking, this approach is very principled and led to much better performance than existing approach that relies on more tricks and/or specific model architectures to work.

**Strength And Weaknesses:**

1. What is the distribution of p(z), the prior used?

2. Is there any convergence guarantee for Alg.1, if $\phi$ is updated via TRNG and $\theta$ via Adam?

3. For Figure 1:
- What is the definition of  z1 and z2 - amplitudes and phases?
- "Right panel: corresponding function samples from our model (blue lines)" - is each line is a sample from q(z)?
- The paper mentioned "all samples pass through the red context example". But from the plot it does not seem so. Is there any guarantee that each sampled functions will certainly pass through the points in the context data points?

4. The paper mentioned "sample a fixed set of auxiliary subtasks" during training. What's the difference between tasks and sub-tasks?

5. The paper mentioned the TRNG-VI require at most first-order gradients of $\tilde{p}(z_l)$. Could you point out which variational update equation needs this gradient info?

**Summary Of The Paper:**

The authors propose a new way of doing Bayesian Meta Learning (BML) by fitting a full-covariance Gaussian Mixture Model (GMM) to approximate each Task Posterior (TP). TRNG-VI proposed by Arenz et al. (2022) was used to ensure efficient and robust optimization of the variational bound. The training complexity is similar to existing approaches, and does amortize inference over tasks and thus does not learn a set encoder architecture. Experiments on synthetic datasets showed the proposed GMM-NP method outperformed all baselines by a large margin over the whole range of context sizes, both in terms of LMLHD and MSE, in different test scenarios regarding task posterior inference, Bayesian optimization, dynamic modeling and image completion.

**Summary Of The Review:**

The GMM-NP approach proposed by this paper achieves much better results than existing approaches for BML, with less requirements in model architectures and objective approximations. The experiments illustrated the effect of each part of the design in a high quality. I would recommend an accept for it, to make people in the BML/NP area aware of the importance of accurate task posterior inference and work along this fundamental direction.

---

> ### Author Response · Authors · 2022-11-15
> **Answer to reviewer 7Kjw (1/2)**
>
> Thanks a lot for your positive review and your helpful comments that allow us to improve our manuscript!
> We gladly answer your remaining questions and update our submission accordingly:
>
> - **What is the distribution of $p(\boldsymbol z)$, the prior used?**:
>
>   As mentioned in Sec. 4, paragraph "Meta-Training", we start from a "randomly initialized variational GMM".
>   Concretely, given a number $K$ of components for the GMM task posterior (TP) $q_{\boldsymbol \phi}(\boldsymbol z)$, we use a prior $p(\boldsymbol z)$ with $K$ components.
>   To initialize the means $\boldsymbol \mu_k$, covariances $\boldsymbol \Sigma_k$, and mixture weights $w_k$ for $k \in \{1, \dots, K \}$, we use the same simple heuristic as Arenz et al., 2022:
>
>   - Draw the means $\boldsymbol \mu_k$ from a $d_z$-dimensional standard Normal distribution,
>   - The covariances $\boldsymbol \Sigma_k$ are initialized as diagonal matrices (with $1$ on the diagonal),
>   - The weights are initialized uniformly as $w_k = 1/K$.
>
>   We add this more detailed description of the initialization procedure in Sec. A.1.2.
>
> - **Is there any convergence guarantee for Alg. 1, if $\boldsymbol \phi$ is updated via TRNG and $\boldsymbol \theta$ via Adam?**:
>
>   Yes, there is a convergence guarantee for our GMM-NP algorithm.
>   Indeed, our algorithm inherits the convergence guarantee of variational Bayes as discussed, e.g., in (Bishop, 2006) or (Murphy, 2022).
>   In particular, convergence is independent of the concrete optimization strategy used for $(\boldsymbol \phi, \boldsymbol \theta)$:
>   as long as both the E-step (step in $\boldsymbol \phi$) and the M-step (step in $\boldsymbol \theta$) increase the ELBO objective, the **algorithm is guaranteed to converge to a local optimum of the ELBO**.
>   While in standard reparametrized variational Bayes (as employed by the SOTA in NP-based BML) $(\boldsymbol \phi, \boldsymbol \theta)$ are optimized jointly using, e.g., Adam, our method alternately performs a step in $\boldsymbol \phi$ using TRNG-VI and a step in $\boldsymbol \theta$ using Adam.
>   Nevertheless, both steps increase the ELBO, so our algorithm will converge.
>
>   We would also like to point you to Sec. 4 of our manuscript, where we explain why our GMM-NP algorithm **improves the convergence behaviour w.r.t. the marginal likelihood** in comparison to existing NP-based BML approaches:
>   Recall that the convergence guarantee of the classical EM algorithm w.r.t. the marginal likelihood is lost as soon as the E-step becomes intractable, i.e., as soon as the posterior distribution cannot be computed exactly, and thus has to be approximated by a variational distribution, cf., e.g., (Bishop, 2006) and (Murphy, 2022).
>   This is the case for most models of reasonable complexity, e.g., for the VAE or the NP.
>   Our GMM-NP model is no exception here, as we build on the NP model for which the task posterior distribution (TP) cannot be computed analytically.
>   Convergence of the marginal likelihood is guaranteed for a tight ELBO, which is the setting of the aforementioned EM algorithm and only the case for a perfect TP approximation, i.e., if $\mathrm{KL} (q_{\boldsymbol \phi}(\boldsymbol z) || p_{\boldsymbol \theta}(\boldsymbol z | \mathcal D^c)) = 0$.
>   For imperfect approximations, the tightness of the bound is controlled by the variational gap $\mathrm{KL} (q_{\boldsymbol \phi}(\boldsymbol z) || p_{\boldsymbol \theta}(\boldsymbol z | \mathcal D^c)) > 0$.
>   A better approximate posterior $q_{\boldsymbol \phi}(\boldsymbol z)$ yields a tighter ELBO, which in turn brings us closer to the EM setting, i.e., typically improves convergence.
>   Our GMM-NP algorithm builds exactly on this insight: we use an expressive TP approximation (full covariance GMM) and a powerful optimizer for $\boldsymbol \phi$ (TRNG-VI) to obtain a tighter ELBO than existing methods in order to achieve optimization of the model parameters in a way that efficiently maximizes the marginal likelihood.
>
>   We add an extended discussion of convergence properties of our algorithm in Sec. A.1.4.

---

> > ### Author Response · Authors · 2022-11-15
> > **Answer to reviewer 7Kjw (2/2)**
> >
> > - **Questions to Figure 1**:
> >
> >   - Your intuition is correct! The model encodes the task descriptors (i.e., amplitudes and phases for the case of the sinusoidal task) in the latent space.
> >     However, the structure of the latent space is learned automatically by the joint optimization procedure for $\boldsymbol \theta$ and $\boldsymbol \phi$ as described in Sec. 4.
> >     Thus, $\boldsymbol z_1$ and $\boldsymbol z_2$ do not necessarily correspond directly to amplitudes and phases, respectively.
> >     Rather, their meaning is given by the learned, nonlinear embedding (defined implicitly by the decoder parameters $\boldsymbol \theta$) of these task descriptors in the latent space.
> >   - Correct, each blue line is corresponds to one sample from $q_{\boldsymbol \phi}(\boldsymbol z)$!
> >   - You are right, it is more accurate to say that all samples pass *close* to the context data point.
> >     Thanks for pointing this out!
> >     We corrected this in the revised version of our manuscript.
> >     The reason for this is that the data contains quite some noise, so there remains ambiguity, which is reflected by the nonvanishing spread of the samples around the context data.
> >     Note that given noisy data this behavior is correct and desirable (we do not want to overfit to the noise).
> >
> > - **What is the difference between tasks and sub-tasks?**:
> >
> >   A *task* with index $\ell$ is defined as the set of available noisy evaluations $\mathcal D_\ell = ( \boldsymbol x_{\ell,1:N}, \boldsymbol y_{\ell, 1:N} )$ drawn from an unknown function $f_\ell$ (cf. Sec. 3.2).
> >   Thus, a task is all data our algorithm has available to learn about $f_\ell$.
> >   A *subtask* of task $\ell$ is an arbitrary subset of $\mathcal D_\ell$.
> >   As described in Sec. 4, paragraph "Meta-Training", in NP-based Bayesian meta-learning, it is standard to train the models not only on the full task $\mathcal D_\ell$, but on randomly sampled subtasks (with varying sizes) in order to encourage the model to produce reasonable predictions for context sets of various sizes.
> >   Thanks for pointing out that our description was unclear!
> >   We improved on that in our revised manuscript, cf. Sec. A.3.2.
> >
> > - **Which variational update equation needs gradient info of $\tilde p_{\ell}(\boldsymbol z_{\ell})$?**:
> >
> >   The gradients are required to compute the quantities $\boldsymbol R_{\ell,k}$ and $\boldsymbol r_{\ell,k}$ in Eqs. (8a) and (8b).
> >   Expressions for these are provided in App. A.1.1. (Eqs. (12a) and (12b)).
> >   Here, the gradients w.r.t. $\boldsymbol z_{\ell}$ of the functions $h_{\ell,k}$ (Eq. (13)) enter, and these functions contain $\tilde p_{\ell}(\boldsymbol z_{\ell})$.
> >
> >
> > We would like to thank you again for your thorough and helpful review!
> > We hope that our comments further convince you of the quality of our submission and of it's relevance to the BML community!
> >
> > Best regards,
> >
> > The authors
> >
> > ---
> >
> > References:
> >
> > Bishop, "Pattern Recognition and Machine Learning", 2006
> >
> > Murphy, "Probabilistic Machine Learning: Advanced Topics", 2022

---

### Official Review · Reviewer_FbJt · 2022-10-25

**Confidence:** 3
**Correctness:** 4
**Technical Novelty And Significance:** 3
**Empirical Novelty And Significance:** 3
**Recommendation:** 8

**Clarity, Quality, Novelty And Reproducibility:**

The paper is clear, easy to read, and the authors provide code for easy reproducibility. To my knowledge, the approach is novel and highlights the importance on the quality of variational approximation when working with variational inference models.

**Strength And Weaknesses:**

Strengths:
- Overall, the paper was clear and easy to follow. The preliminaries introduced the intuition behind the core topics of the approach well. The description of VI was clear and made the exposition of the core contribution of this paper easy to understand.
- The approach is simple yet effective, and serves to highlight the utility of more expressive task posteriors in a latent-space meta-learning context.
- The experiments are thorough and demonstrate the capabilities of the architecture at representing an accurate task posterior both qualitatively, as well as quantitatively via the log-marginal likelihood metric on synthetic task distributions as well as the utility of accurate posterior modeling on downstream tasks such as bayesian optimization, dynamics modeling, and image completion. I also appreciated the ablations measuring ELBO looseness as a function of VI strategy and distribution expressivity to support the author's hypothesis that the quality of variational approximation correlated with downstream performance.

Weaknesses:
- As stated by the authors, a main limitation is the added computational complexity due to the non-amortized variational inference. While this leads to higher quality task posteriors, it brings   a natural tradeoff of prediction quality vs runtime. It would have been useful to see more details on the computational overhead in terms of wall clock speed -- how many iterations of VI are necessary for convergence on average? How does wall clock runtime compare to algorithms with more complex encoders, such as attention based NPs?
- The method requires optimizing a separate task posterior not only for each task in the meta-dataset, but also for each context size size considered for each task, each requiring storing the set of GMM parameters. In contrast, an amortized inference approach would share parameters across tasks, but here the parameters would be the weights of an encoding network. How do the number of training parameters compare across methods?
The number of latent-dimensions differed between algorithms (and was selected in the hyperparameter optimization). However, this made it difficult to compare different approaches -- perhaps in a higher dimensional latent space, a unimodal distribution can encode similar information as in a multimodal distribution in a lower-dimensional latent space? The paper would be strengthened by either a comparison between methods with a fixed latent dimension, or a more in-depth discussion of any trends in the choice of latent-space dimension that emerge through the hyperparameter optimization.

Questions:
- What data was used for hyperparameter optimization?

**Summary Of The Paper:**

This paper proposes a method for Bayesian meta-learning that aims to learn a conditional generative model f(x,z) such that variational inference on a latent z can model the posterior predictive distribution given context data from a new task. Specifically, the authors draw connections between their approach and the Neural Process (NP) model, and emphasize how performing non-amortized variational inference via trust-region natural gradient optimization of a GMM distribution on z can yield higher quality posterior predictive distributions than NP-style models, even those with more complex encoders.

**Summary Of The Review:**

Overall, I think this is a strong paper, presenting a simple, novel algorithm which achieves impressive results. The work is well presented, and the experimental evaluation is thorough, so I would recommend acceptance.

---

> ### Author Response · Authors · 2022-11-15
> **Answer to reviewer FbJt**
>
> Thank you very much for your effort in reviewing our paper.
> We are grateful for your positive and detailed remarks!
> We provide answers to your remaining questions and improve our manuscript based on your comments:
>
> - **We provide a more detailed analysis of computational complexity**:
>   - **Measuring wall-clock runtimes**:
>     As you suggested, we provide a runtime comparison in Sec. A.5.6 of the updated manuscript.
>     We compare **learning curves of all algorithms for meta-training**, supporting our claim that GMM-NP incurs a computational cost (in terms of wall-clock runtime) comparable to the baselines during meta-training.
>     We furthermore provide the **learning curves for fitting variational GMMs at test time** using our GMM-NP algorithm (as suggested, both in terms wall-clock runtime and TRNG-VI optimization steps).
>     Depending on the context size, our algorithm requires approximately 0.1s - 1s per test task or 100-500 steps to converge.
>
>   - **Comparison of HPO results**:
>     Following your remarks, we provide the **results and a discussion of the hyperparameter optimization of the latent dimensionality and the number of model parameters** of the compared methods in Sec. A.5.7 of the updated manuscript.
>     We find that your intuition indeed is correct: the baseline methods tend to favour higher-dimensional latent spaces.
>     Likewise, we observe that while the total number of variational parameters of GMM-NP is naturally higher in comparison to the baselines during meta-training (as GMM-NP does not use amortization), our expressive TP approximation allows comparably lightweight decoder architectures.
>     This is again in accordance with your intuition: expressive decoders are required to transform simple TP approximations into expressive predictive distributions.
>     Note further that the variational parameters belonging to different tasks are not coupled in non-amortizing architectures such as ours, which allows trivial parallelization of the variational optimization between tasks, explaining why the computational cost is easily managable (cf. the discussion above).
>
> - **What data was used for hyperparameter optimization?**:
>
>   To optimize the hyperparameters of each model on a given experiment, we trained the model on the meta-data and evaluate the LMLHD metric on unseen validation data after convergence.
>   We perform 256 Bayesian optimization trials to maximize the validation LMLHD metric for each model-dataset combination.
>   Afterwards, we report the results of 8 random seeds retrained with the best found parameter configuration on the test data.
>
> Thank you again for your valuable review!
> We hope that the additional experimental results and our comments further convince you of the quality of our work and of it's relevance for the BML community!
>
> Best regards,
>
> The authors

---

> > ### Comment · Reviewer_FbJt · 2022-11-29
> > **Thanks for the updates**
> >
> > The updates strengthen the paper and address my concerns. I remain an advocate for the paper's acceptance.

---

### Official Review · Reviewer_sb3h · 2022-10-25

**Confidence:** 2
**Correctness:** 3
**Technical Novelty And Significance:** 3
**Empirical Novelty And Significance:** 3
**Recommendation:** 6

**Clarity, Quality, Novelty And Reproducibility:**

Clarity&quality: this paper is well-written
novelty: this paper solves a critical problem and has good novelty
reproducibility: full experimental setting and release github code repo

**Strength And Weaknesses:**

Strengths:
1. identifies the optimization suboptimal and solves by combining VI with natural gradients and trust regions
2. strong experimental results which covers multiple settings and applications
Weakness/Questions:
1. As said in the paper, the additional overhead in the testing time can be parallelized. Can you give some empirical results to support it?
2. How would your proposed method be used for Bayesian uncertainty learning models, especially those multi-modal, time-varying models for classification/regression?

**Summary Of The Paper:**

This paper proposes GMM-NP, a novel BML algorithm inspired by the NP model architecture, which focuses on accurate task posterior inference. Despite its simplicity, GMM-NP outperforms the state-of-the-art on a range of experiments and demonstrates its applicability in practical settings

**Summary Of The Review:**

This paper is well-done for improving the optimization and inference of BNP. This paper can be improved by providing more empirical results and elaborations.

---

> ### Author Response · Authors · 2022-11-15
> **Answer to reviewer sb3h**
>
> Thank you very much for your review and for your generally positive remarks!
> We gladly answer your remaining questions below and adapt our manuscript according to your comments:
>
> - **Empirical results for computational overhead**:
>
>   Following your and reviewer FbJt's remarks, we provide a runtime comparison in Sec. A.5.6 of the updated manuscript.
>   We compare **learning curves of all algorithms for meta-training**, supporting our claim that GMM-NP incurs a computational cost (in terms of wall-clock runtime) comparable to the baselines during meta-training.
>   We furthermore provide the **learning curves for fitting variational GMMs at test time** using our GMM-NP algorithm.
>   Note that our implementation exploits the fact that the computations can be trivially parallelized over tasks and, depending on the context size, requires approximately 0.1s - 1s per test task or 100-500 steps to converge.
>
> - **Application to Bayesian uncertainty learning models (multi-modal, time-varying) for classification/regression**:
>
>   - **Comparison to single-task Bayesian models**: In comparison to single-task Bayesian models such as Bayesian neural networks (BNNs), which compute a Bayesian belief over model parameters, we study the multi-task NP-based Bayesian meta-learning (BML) setting and, thus, encode Bayesian uncertainty in a latent variable which is fed as an input to the decoder DNN.
>   In comparison to, e.g., BNNs, this setting has the advantage that the latent variable is comparably low-dimensional, which allows much more accurate inference schemes such as our proposed GMM-NP using TRNG-VI, enabling multi-modal TP approximations.
>
>   - **Application to classification**: While we only studied regression experiments in our submission, our model is not limited to this case and the extension to classification is straightforward (by just changing the likelihood defined by the decoder DNN accordingly).
>   In particular, it does not touch any of the novel design choices w.r.t. TP inference we presented in our work.
>
>   - **Application to time-varying data**: Our approach can be easily applied to settings where the context data changes over time.
>   Indeed, all that is necessary is to adapt the variational GMM to the updated context data set.
>   As discussed in Sec. 4, in such settings it would be natural to warmstart the TP optimization with the current TP approximation to speed up convergence.
>
> We would like to thank you again for your effort in reviewing our paper!
> We hope that our extended experimental evaluation and our clarifications convince you of the quality and of the relevance of our submission for the BML community!
>
> Best regards,
>
> The authors

---

### Official Review · Reviewer_tZEh · 2022-10-25

**Confidence:** 4
**Correctness:** 3
**Technical Novelty And Significance:** 3
**Empirical Novelty And Significance:** 3
**Recommendation:** 6

**Clarity, Quality, Novelty And Reproducibility:**

Clarity: The submission is easy to follow.

Quality: All equations and derivations seems correct although I did not check it carefully.

Novelty: The contribution is incremental.

Reproducibility: I did not check it.

**Strength And Weaknesses:**

Strength: The paper is generally easy to follow, and the derivation seems correct to me although I did not check it carefully. Replacing the Gaussian approximated posterior parameterized by an encoder with a Gaussain mixture distribution is an interesting idea.

Weakness: The proposed inference method is inherited from the TRNG-VI in [Arenz et al. 2022] and the contribution is incremental.

**Summary Of The Paper:**

The submission modified the Neural Process model by replacing the Gaussian approximated posterior parameterized by encoder with a Gaussain mixture distribution parameterized by full-covariance matrix. For inference, the proposed method divide the VI into two alternating updating steps between variaitonal and model parameters. For variaitonal parameters, the method uses the existing trust region natural gradient based VI in [Arenz et al. 2022]; for model parameters, it uses the standard VI. The propsoed model and inference is validated on a range of experiments.



**Summary Of The Review:**

From the technical perspective, the methodology in the submission is correct in my eyes (I did not check it carefully) and the experiments are rich and solid. Generally speaking, the submission is a good paper without obvious defects. However, from the model inference perspective, this work simply applied the method in "A Unified Perspective on Natural Gradient Variational Inference with Gaussian Mixture Models" [Arenz et al. 2022] to modify the Neural process model; all contributions mentioned in this work, "full covariance matrix", "Gaussian mixture distribution", "non-amortized", "trust region natural gradient descent", are inherited from this previous work. Therefore, the incremental contribution dampens my enthusiasm to give a very high score.

---

> ### Author Response · Authors · 2022-11-15
> **Answer to reviewer tZEh**
>
> Thank you very much for your effort in reviewing our paper and for your generally positive remarks!
> We would like to comment on your concerns regarding the novelty of our method.
>
> Our approach makes use of the inference method TRNG-VI proposed by Arenz et al., 2022.
> As you correctly pointed out, this method is not new, but we applied it in a novel context (in a **multi-task setting**, i.e., to infer the task posterior (TP) of Bayesian meta-learning (BML), as opposed to the single-task setting studied by Arenz et al., 2022).
> While we believe that this in itself is a significantly novel contribution, we would like to highlight that our submission goes way beyond just proposing the resulting GMM-NP algorithm.
> In particular, we are confident that the following **conceptual and empirical contributions** are both novel and of high relevance for the BML community:
>
> - We **rigorously demonstrate the need for expressive TP approximations and accurate TP inference schemes** by identifying that the current practice of amortized, factorized, unimodal Gaussian TPs for BML with standard reparametrized gradients is suboptimal and eventually leads to ineffective optimization of the model parameters (Sec. 4).
> - We support this claim by **extensive empirical evidence** (Sec. 5.2), highlighting that amortization and inaccurate TP approximations indeed lead to loose ELBOs, harming optimization performance.
> We believe that the BML community is currently not aware of this fact, and that our paper is thus of high relevance for further research in BML.
> - Our algorithm demonstrates that **accurate TP approximations are tractable and efficient** for BML, e.g., by using TRNG-VI (Arenz et al., 2022) (Sec. 4), and lead to **markedly improved performance**.
> - We employ a **rigorous, fair, and reproducible evaluation methodology** (Sec. 5) to support this claim, and compare a wide range of SOTA BML algorithms on a wide range of practically relevant BML experiments.
> - Most notably, our analysis and our results demonstrate that **accurate TP inference allows accurate BML with conceptually much simpler architectures compared to the state of the art**.
> For example, we show that the widespread approach of adding deterministic computation paths actually harms epistemic uncertainty estimation, and that accurate TP inference should be considered as a more promising research avenue for BML.
>
> As also pointed out by the other reviewers, we are confident that our paper is "**novel and highlights the importance on the quality of variational approximation**" (reviewer FbJt) and that "**people in the BML/NP area [should be made] aware of [... the] work along this fundamental direction**" (reviewer 7Kjw).
>
> Thank you again for allocating your time to review our paper!
> We hope that our remarks can convince you of the novelty of our submission and of it's relevance for the community!
>
> Best regards,
>
> The authors
>
> ---
>
> References:
>
> Arenz et al., "A Unified Perspective on Natural Gradient Variational Inference with Gaussian Mixture Models", 2022

---

### Author Response · Authors · 2022-11-15
**General answer to all reviewers**

Dear reviewers,

we would like to thank you for you effort in reviewing our paper! We are very grateful for your generally positive remarks, judging our submission to be "a good paper" (reviewer tZEh) which is "novel and highlights the quality of variational approximation" (reviewer FbJt) for Bayesian meta-learning, with "rich and solid experiments" (reviewer tZEh) that "illustrate the effect of each part of the design in a high quality" (reviewer 7Kjw), leading to results that "people in the BML/NP area [should be made] aware of" (reviewer 7Kjw).
Thank you also for your helpful questions, comments, and suggestions, which we address in separate answers below.
We also provide an updated version of our manuscript where we mark additions in blue.

Best regards,

The authors

---

### Decision · Program_Chairs · 2023-01-20

**Decision:**

Accept: poster

**Justification For Why Not Higher Score:**

Paper makes a solid contribution, but probably doesn't warrant a spotlight.

**Justification For Why Not Lower Score:**

Reviewers all voted to accept.

**Metareview: Summary, Strengths And Weaknesses:**

The authors introduce a a new Bayesian meta-learning approach, GMM-NP. It uses an existing inference method, TRNG-VI in a new setting, building on prior work on neural processes. The paper's value lies in its convincing empirical argument that the inaccurate posterior approximations built on simple models (factorized Gaussian) lead to suboptimal performance. The reviewers praised the paper's clarity and reproducibility, while noting that it is some ways incremental. The authors plan to address concerns about added overhead in terms of runtime.

**Note From Pc:**

if the above contains the word "oral" or "spotlight" please see: "oral" presentation means -> notable-top-5% and "spotlight" means -> notable-top-25%. As stated in our emails, we are disassociating presentation type from AC recommendations